# Context-specific regulatory genetic variation in *MTOR* dampens neutrophil-T cell crosstalk in pneumonia-associated sepsis

Ping Zhang [1,2] ✉, Patrick MacLean [2], Alicia Jia [2], Callum R. O'Neill [2], Alice Allcock[2], Ethan Prince [2], Bora Ozcan [3], Roman M. Doll [3], Imogen Dyne[2], Kiki Cano-Gamez [2,4], Hanyu Qin[1,2], Chloe Wainwright[2], Giuseppe Scozzafava[2], Andrew C. Brown [2], James O. J. Davies [3,5], Amanda Y. Chong [2], Alexander J. Mentzer [1,2], Katie L. Burnham [6], Emma E. Davenport[6] & Julian C. Knight [1,2,5] ✉

Sepsis is a heterogeneous clinical syndrome with a high mortality, requiring personalised stratification strategies. Here, we characterise genetic variation that modulates *MTOR*, a critical regulator of metabolism and immune responses in sepsis. The effects are context specific, involving a regulatory element that affects *MTOR* expression in activated T cells with opposite effect in neutrophils. We show that the G-allele of the lead variant, rs4845987, which is associated with decreased risk of type 2 diabetes, reduces *MTOR* expression in T cells and improves survival in sepsis due to pneumonia, with effects specific to sepsis endotype. Using ex vivo models, we demonstrate that activated T cells promote immunosuppressive neutrophils through released cytokines, a process dampened by hypoxia and the mTOR inhibitor rapamycin. Our work demonstrates an epigenetic mechanism fine-tuning *MTOR* transcription and T cell activity via the variant-containing regulatory element, which further exhibits an allelic effect upon vitamin C treatment. These findings reveal how genetic variation interacts with disease state to modulate immune cell-cell communication, providing a framework for stratified therapy in sepsis.

Sepsis is a heterogeneous clinical syndrome characterised by life-threatening organ dysfunction associated with a dysregulated host response to infection. This includes, in a patient and disease state-specific manner, decreased lymphopoiesis and adaptive immunity, sustained inflammation, and myeloid pathologies[1]. The mechanistic basis for these diverse features and individual risk, which impact not only acute illness but also poor long-term clinical outcomes in survivors, remains unresolved[1,2]. To progress the goal of tackling heterogeneity in sepsis and advance personalised stratification strategies for successful targeted therapeutics, subtypes of sepsis defined by clinical features and white blood cell gene expression have been reported[3–6]. For example, a specific transcriptomic sepsis response signature, SRS1, identifies a subset of patients with more severe disease involving T cell exhaustion, immunosuppressive neutrophils, hypoxia

[1]Chinese Academy of Medical Sciences Oxford Institute, Nuffield Department of Medicine, University of Oxford, Oxford, UK. [2]Centre for Human Genetics, Nuffield Department of Medicine, University of Oxford, Oxford, UK. [3]MRC Molecular Haematology Unit, MRC Weatherall Institute of Molecular Medicine, Radcliffe Department of Medicine, University of Oxford, Oxford, UK. [4]Department of Clinical and Biomedical Sciences, University of Exeter, Exeter, UK. [5]Oxford National Institute of Health Research Biomedical Research Centre, University of Oxford, Oxford, UK. [6]Wellcome Sanger Institute, Wellcome Genome Campus, Hinxton, Cambridge, UK. ✉e-mail: ping.zhang@well.ox.ac.uk; julian.knight@well.ox.ac.uk

and glycolysis networks[3,7], informing a more stratified approach to understanding specific maladaptive sepsis responses[8–10].

The dysregulated host responses in sepsis span innate and adaptive immunity, and neutrophil-to-lymphocyte ratio (NLR) has been proposed as a biomarker that reflects this interplay. Dysregulation of NLR is observed in many specific infectious diseases, as well as autoimmune disorders and cancer, providing insights into susceptibility and pathogenesis[11,12]. A high neutrophil-to-T cell ratio is correlated with poor prognosis and increased mortality in sepsis[13] and COVID-19 patients[14]. As one of the critical modulators of adaptive immunity, neutrophils can regulate the adaptive immune response either by releasing antimicrobial peptides and cytokines, or by direct cell-surface interactions[15,16]. The complex role of neutrophil subsets in immunostimulation versus immunosuppression is not fully understood. Antigen-presenting aged neutrophils (APANs) drive CD4+ T cell proliferation, exacerbating inflammation in sepsis[17], while immunosuppressive neutrophils inhibit the proliferation and activation of CD4+ T cells and are prone to detrimental NETosis during severe sepsis[8]. However, the reciprocal interaction between neutrophils and T cells, and the role of neutrophils as potential effectors in the context of sepsis, remain incompletely defined.

Recent studies have identified human expression quantitative trait loci (eQTLs) as important genetic contributors to context-dependent phenotypic variation[18–20]. While most currently reported eQTLs exhibit consistent effects across various contexts, those associated with disease may display cell-type specificity and can be linked to cellular dysregulation[21,22]. The knowledge gained from emerging evidence revealing differences in chromatin accessibility, histone modifications and transcription factor binding across cell types can be harnessed to prioritise causal eQTLs and potential drug targets[23,24]. However, understanding the mechanisms by which eQTLs affect gene expression, and how the cellular environment interacts to create cell-type specific effects, remains unresolved.

Driven by the potential contribution of genetic factors to the observed cellular dysregulation in sepsis[3,25], here we investigate the mechanistic basis of a context-specific eQTL involving *MTOR*. MTOR is a key metabolic regulator, controlling a signalling hub that integrates immune receptor and metabolic signals including the switch to pro-inflammatory glycolysis in sepsis[26]. We have previously reported that *MTOR* is downregulated in SRS1 patients, and that there was an eQTL at this locus whose effect was specific to patients with the SRS1 endotype[3]. We now map this association to a genetic variant that we find residing in a context-specific regulatory element, and show that this has an inverted effect on *MTOR* expression between activated T cells and neutrophils, decreasing and increasing expression, respectively. We demonstrate how an eQTL can modulate cell-to-cell communication, here in sepsis through a negative feedback loop between activated T cells and detrimental neutrophils, mediated by mTOR activity and hypoxia. The G-allele of this variant is associated with reduced mTOR signalling and cytokine release in T cells, leading to improved survival for sepsis patients. Furthermore, we characterise the epigenetic mechanisms regulating *MTOR* transcription in resting and activated T cells, providing insights into the survival of memory T cells and the detrimental effects of T cell activation during sepsis.

## Results

### A sepsis *MTOR* eQTL involves directional association between neutrophils and activated T cells

We first sought to understand the cellular context and specificity of the *MTOR* eQTL showing association dependent on sepsis endotype[3]. We fine mapped the association in patients from the UK Genomic Advances in Sepsis (GAinS) cohort using total white blood cell bulk RNA-seq and SNP genotyping (*n* = 823 samples from 638 sepsis patients)[25]. This localised the eQTL to a haplotype block with 25 variants in strong linkage disequilibrium (LD) ($r^2 > 0.95$), tagged by

rs4845987 (Fig. 1a and Supplementary Fig. 1a, b). The minor G-allele of rs4845987 is associated with increased expression of *MTOR* in whole blood and this is magnified in SRS1 patients compared to non-SRS1 patients (Fig. 1b).

To explore how this observation of endotype specificity may relate to the cellular specificity of the *MTOR* eQTL, we performed cell-type deconvolution of the total white blood cell bulk RNA-seq. This showed differences specific to SRS endotype involving both innate and adaptive cell populations, specifically neutrophil and T cells subsets (Fig. 1c). We therefore considered the relationship of the eQTL with NLR, noting that higher NLR based on hospital laboratory leucocyte differential counts from our previously published cohorts[27,28] shows association with more severe all-cause sepsis, COVID-19 and post-operative pneumonia (Supplementary Fig. 1c–e); and a positive correlation with the quantitative transcriptomic sepsis response score (SRSq) for the likelihood of SRS1 endotype[7] (Supplementary Fig. 1f). We found that the *MTOR* eQTL tagged by rs4845987 shows interaction with NLR, with increased expression of *MTOR* for the G allele in patients with heightened NLR (compared to those with lower NLR) (Fig. 1d). We then asked how often this relationship may be observed. From our whole-blood sepsis eQTL data[25], we found that 565 out of 12,726 genome-wide independent eQTL effects in sepsis show interaction with NLR (FDR < 0.01, Supplementary Table 1), and 65% (365 out of 565) of these also showed significant interaction (FDR < 0.01) with SRS endotype (Supplementary Fig. 1g), with strong concordance between SRS and NLR interaction effects (Pearson's *r* = 0.95; Supplementary Fig. 1h). This demonstrates that there are likely to be multiple independent genetic modulators of immune-relevant phenotypes dependent on relative abundance of neutrophil and lymphocyte populations, increasing the potential for clinical relevance.

To identify immune cell types and treatment settings linked to the *MTOR* association, we next performed colocalisation analysis using data curated by the eQTL Catalogue and GTEx consortium[29,30]. Among 127 datasets from 75 tissues/cell types and 14 treatments, 11 different cellular or tissue contexts demonstrated shared genetic associations for the lead eQTL with *MTOR* expression, including T cells stimulated with anti-CD3/CD28 beads and neutrophils (PP4 > 0.95; Fig. 1e, g and Supplementary Table 2), but not resting T cells (Fig. 1h). We found that the minor G-allele effect of rs4845987 on *MTOR* expression was inverted between activated T cells and neutrophils, decreasing and increasing expression respectively (Fig. 1f). This aligns with our interaction analysis showing lower *MTOR* expression with possession of the G-allele in non-SRS1 and low NLR patients (Fig. 1b, d), where T cell function was less attenuated compared to SRS1 and high NLR patients (phenotypes associated with more severe disease). By contrast, no effect was seen for monocyte-to-lymphocyte ratio (MLR) (Supplementary Fig. 1i).

### The *MTOR* allele associated with reduced T cell activity is protective for survival during sepsis

We next investigated whether there was evidence of association with outcome in sepsis for the *MTOR* eQTL lead SNP. In the UK GAinS cohort, we found that patients with a copy of the minor G-allele exhibited significantly reduced 28-day mortality compared to those with the major C-allele (*p* = 0.0010, HR = 0.60 [95% CI 0.44–0.81]; Fig. 2a; *n* = 737). This association was restricted to patients with sepsis due to community-acquired pneumonia (CAP) (Fig. 2a) and not observed in non-CAP sepsis patients (*p* = 0.45; Supplementary Fig. 2a; *n* = 384). In an independent Sepsis Immunomics (SI) cohort[8], we genotyped the sepsis CAP patients for rs4845987 (see "Methods"). Consistent with UK GAinS, the G allele was associated with improved sepsis survival (*p* = 0.031, HR = 0.32 [95% CI 0.11–0.90]; Fig. 2b; *n* = 102). A similar protective effect was observed in a further independent validation cohort from the UK Biobank for participants who had

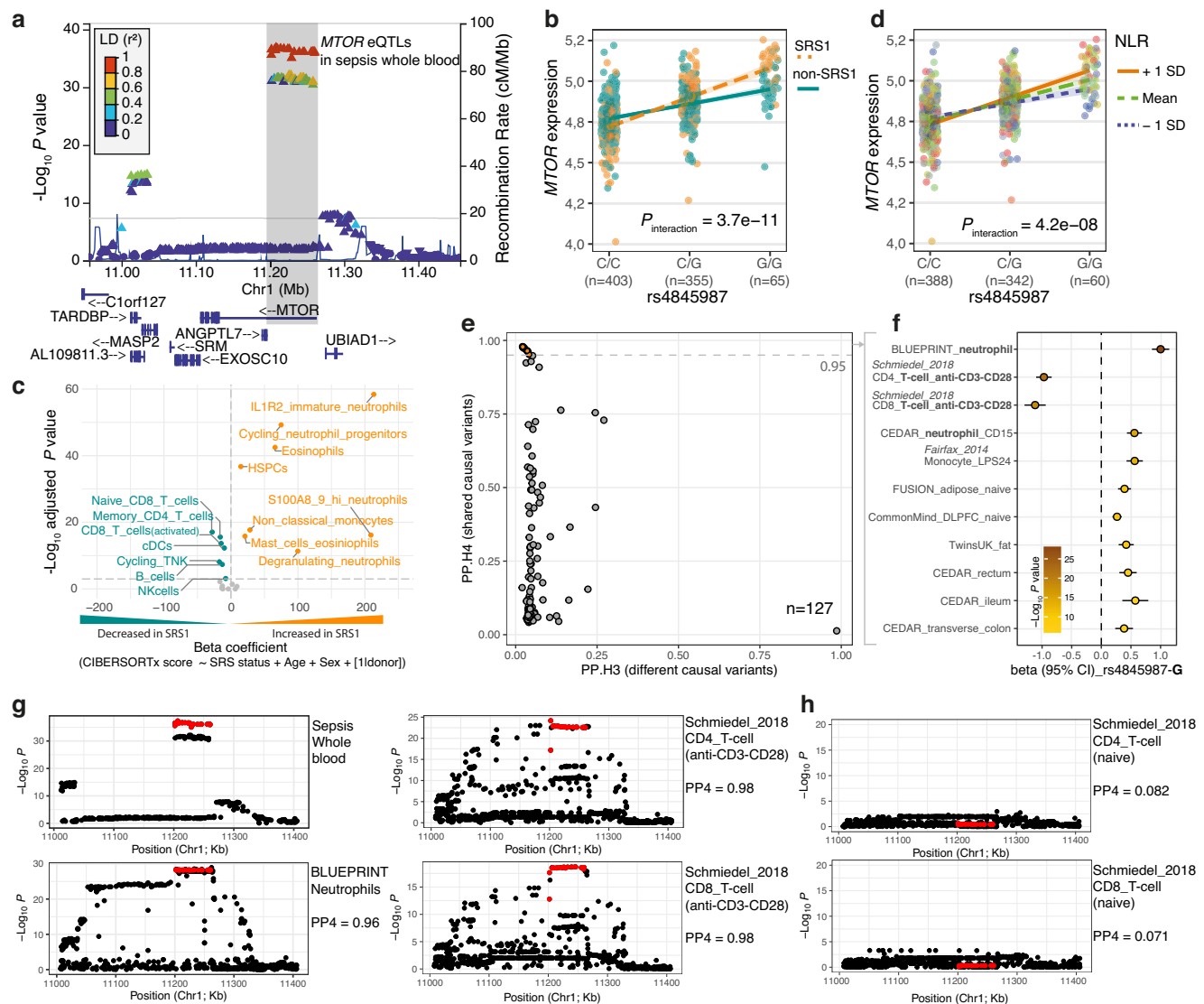

**Fig. 1 | The intronic *MTOR* eQTL interacts with SRS endotypes and shows inverted association between neutrophils and activated T cells. a** Regional plot illustrating *MTOR* lead eQTL variants (in red) identified in whole blood of sepsis patients. **b** The interaction effects of SRS endotypes on the *MTOR* eQTL association in whole blood. **c** Volcano plot depicting the differential cell abundance in SRS1 sepsis patients relative to the non-SRS1 patients from the GAinS cohort. The absolute proportion of each cell type (CIBERSORTx absolute scores) was obtained from deconvolution of bulk RNA-seq using a single-cell RNA-seq reference panel derived from sepsis patients[8]. *p* value was calculated by linear mixed model adjusted for age and sex. The horizontal dashed line represents the Bonferroni-corrected *p* value 0.001. **d** The interaction effects of NLR on the *MTOR* eQTL

association in whole blood. **e** Colocalisation of the *MTOR* eQTL association from sepsis with 127 datasets derived from normal tissue/cell types with different treatment conditions. Summary statistics for the eQTL datasets were retrieved from *eQTL Catalogue*[29]. Posterior probabilities of shared (PP.H4) or distinct (PP.H3) causal variants were computed using R package *coloc*[69]. **f** Effect size and nominal *p* value for the lead eQTL across the top 11 colocalised datasets as highlighted in **e** in orange with PP4 ≥ 0.95. **g, h** Regional association plots for the *MTOR* eQTL in sepsis whole blood, neutrophils, activated T cells and naïve T cells derived from healthy donors. The lead eQTL variants identified in sepsis are highlighted in red. PP4 was calculated using *coloc* (see "Methods"). See also Supplementary Figs. 1 and 10.

confirmed bacterial pneumonia (*p* = 0.0041, HR = 0.68 [95% CI 0.52–0.88]; Fig. 2c; *n* = 1125; see "Methods").

To further investigate the interplay between cellular features and allelic differences in clinical outcomes, we stratified the GAinS cohort into groups based on SRS status (defined as SRS-latest [assigned based on samples from the latest available time point for each patient]; SRS1-ever [assigned if any SRS1 sample was detected across time points for a given patient]) and NLR levels. We found that in non-SRS1 patients, those with the G-allele have a significantly better survival using logistic regression compared to the SRS1 patient group (OR = 0.43, *p* = 0.0012 when considering SRS_latest; OR = 0.38, *p* = 5.5e−04 using SRS1_ever; Fig. 2d). A stronger protective association of the G-allele was also observed in patients with low NLR compared to high NLR (OR 0.46 vs.

0.51; Fig. 2d). This was not seen with MLR (Supplementary Fig. 2b). Consistent with the immune paresis nature of the SRS1 endotype, the protective effect of the G-allele in UK Biobank patients (Fig. 2c) was observed only in those without an immunosuppressed status (IS) or without cancers (Fig. 2e and Supplementary Fig. 2c, d; see "Methods"). Stratification by T2D and HbA1c (a T2D marker; Supplementary Fig. 2e) further showed that this protective effect was lost in T2D patients and in those with higher HbA1c levels (Supplementary Fig. 2c, d). In line with this, a recent study demonstrated that sustained hyperglycaemia, as seen in T2D, can impair T cell signalling and metabolism[31].

Prolonged hypoxia significantly contributes to multi-organ failure, a hallmark of severe sepsis, and is relevant to a potential role for genetic variants of *MTOR*. We therefore analysed patients requiring a

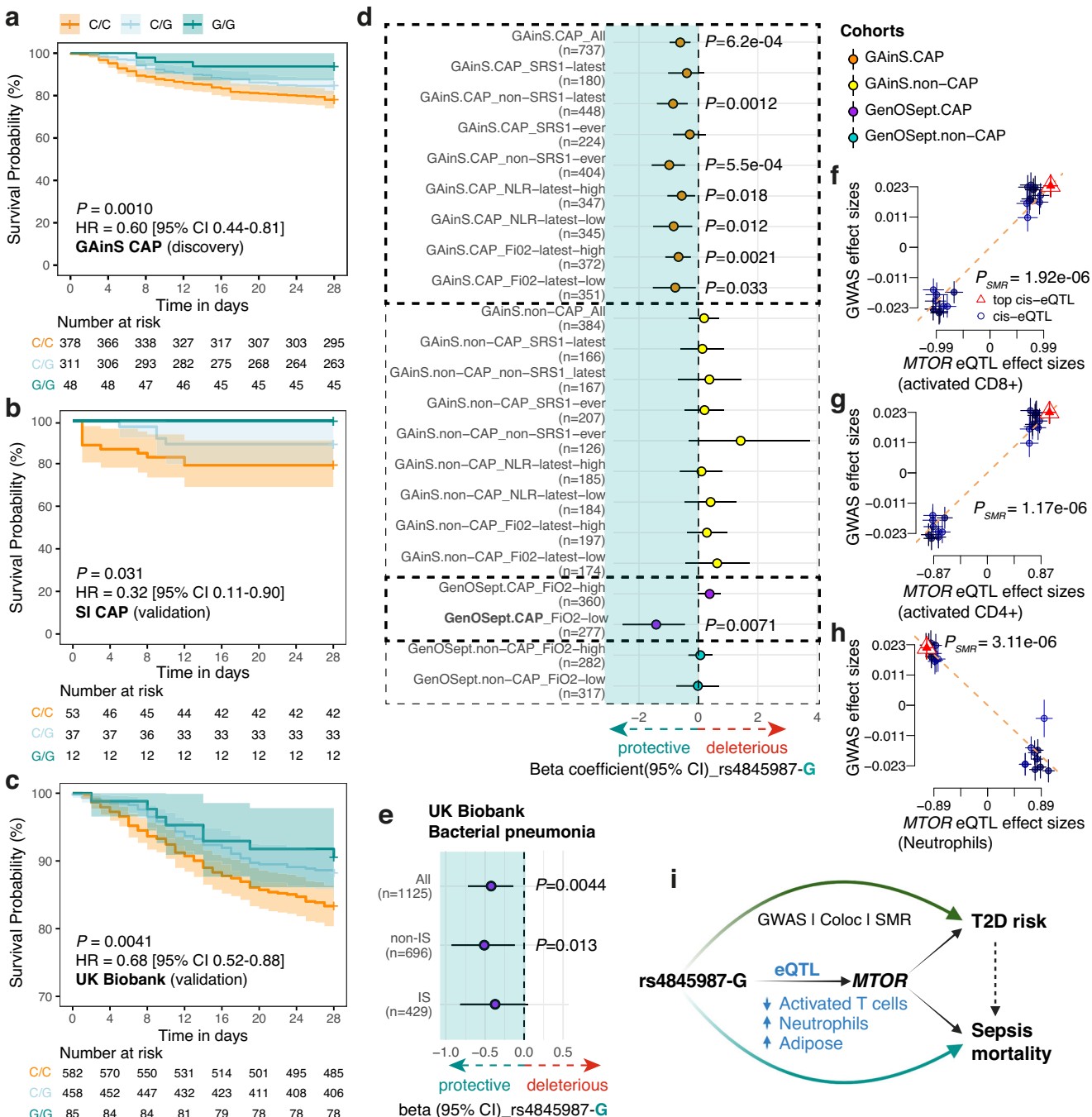

**Fig. 2 | The G-allele associated with reduced T cell activity is protective for survival during sepsis and reduces type 2 diabetes risk.** Kaplan–Meier curves for 28-day mortality in sepsis CAP patients of GAinS (**a**; n = 737), SI (**b**; n = 102) and UK Biobank (**c**; n = 1125) cohorts carrying different alleles of the *MTOR* eQTL rs4845987. Shaded error bands represent the 95% confidence intervals (CI). p value was calculated using the cox regression adjusted for age and gender. **d** Forest plot showing 28-day mortality for GAinS or GenOSept cohort patients carrying the G alleles, stratified according to SRS status, NLR, or FiO2 levels. Beta coefficient and p value were calculated using additive logistic regression, adjusting for age, gender and the first seven genotype principal components from Europeans. Error bars indicate the 95% CI of the beta coefficients. **e** Forest plot showing 28-day mortality in the UKB pneumonia cohort, stratified by immunosuppressed status (IS). Beta coefficients and p values were calculated as described above. Dot plots showing the effect sizes of SNPs used for the SMR/HEIDI analysis from T2D GWAS[37] on the y axis, against eQTLs in activated CD8+ (**f**), activated CD4 + T cells (**g**)[20] and neutrophils (**h**)[71] on the x axis. Error bars represent the standard errors of the SNP effects. **i** Summary plot illustrating the pleiotropic effects of rs4845987-G on reducing T2D risk and sepsis mortality via its allelic effect on *MTOR* expression. See also Supplementary Figs. 2 and 3.

high fraction of inspired oxygen (FiO$_2 \geq$ 0.4) or with a low ratio of arterial oxygen tension (PaO$_2$) to FiO$_2$ (0 < PaO2/FiO2 ≤ 100) as indicators of hypoxaemia during sepsis[32,33]. As expected, we observed high FiO$_2$ and low PaO$_2$/FiO$_2$ were associated with poor survival in the GAinS cohort (HR = 2.6 and 3.8 respectively; p < 0.0001; Supplementary

Fig. S2f). Significant association for FiO$_2$ level, but not PaO$_2$/FiO$_2$, was also observed in another independent cohort GenOSept[34] (Supplementary Fig. S2f). Consistently, in patients with low FiO$_2$ across both cohorts, we observed stronger allelic differences in survival (OR = 0.46, p = 0.033 for GAinS; OR = 0.24, p = 0.0071 for GenOSept; Fig. 2d). We

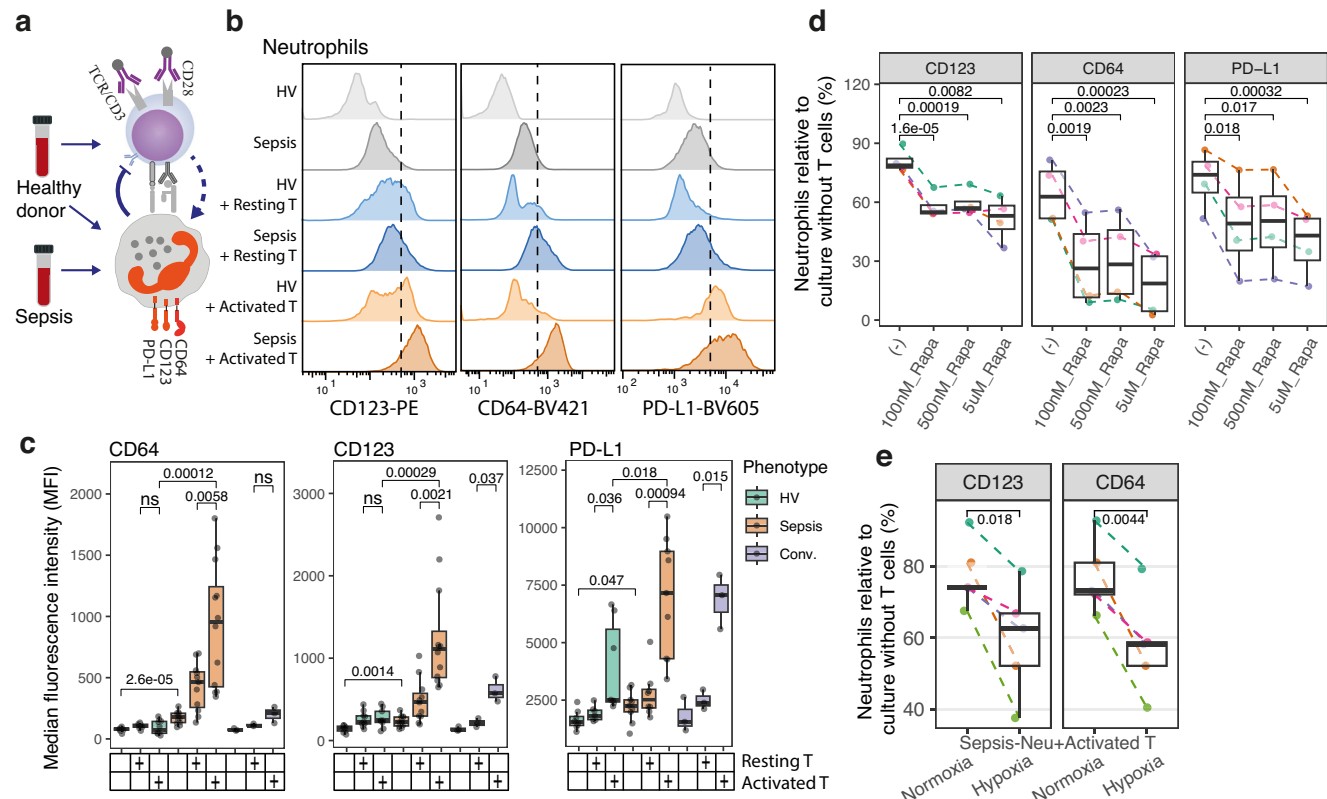

**Fig. 3 | Neutrophil activation induced by T cells was dampened under hypoxia and by mTOR inhibition. a** Schematic of ex vivo co-culture for CD66b+ neutrophil and allogeneic T cells from healthy donors in the absence or presence of anti-CD3/CD28 Dynabeads. **b** Flow cytometry analysis showing the expression of sepsis surface makers on neutrophils derived from sepsis patients and healthy volunteers (HV) after 2 days of co-culture. **c** Box plots showing the median fluorescence intensity (MFI) values measured by flow cytometry. Each dot represents an individual ($n = 12$ sepsis patients; $n = 3$ Conv.; $n = 11$ HVs). Conv. convalescent. $p$ value was calculated by two-tailed $t$-test. ns not significant. Box plots showing the relative expression of markers on sepsis neutrophils co-cultured with activated T cells in the presence of rapamycin ($n = 4$) (**d**), or under hypoxia (1.5% $O_2$) vs. normoxia condition ($n = 5$) (**e**). $p$ value was calculated by two-tailed paired $t$-test. Source data for **c**–**e** are provided as a Source Data file. See also Supplementary Figs. 4 and 5.

did not observe significant associations between the eQTL and survival in non-CAP patients with or without any stratification (Fig. 2d yellow and cyan dots).

## Pleiotropic effects of genetic variant on T2D risk and sepsis survival through modulation of *MTOR* expression

We next further investigated the relationship between the *MTOR* sepsis-associated eQTL and other disease-related traits using the GWAS Catalogue[35]. We found that rs4845987 or SNPs in high LD ($r^2 > 0.8$) were significantly associated ($p \leq 5e{-}08$) with 18 GWAS traits, 11 of which were related to T2D or its associated metabolic traits such as body mass index and insulin resistance (Supplementary Table 3). We observed strong colocalisation between the *MTOR* eQTLs and risk variants for T2D in two GWAS studies[36,37] (PP4 = 0.75 and 0.97, respectively; Supplementary Fig. 3a, b). The minor G-allele of rs4845987, which is protective for sepsis survival, was also associated with a reduced risk of T2D (Supplementary Fig. 3a). Individuals with T2D have an increased risk of developing sepsis, and sodium-glucose cotransporter-2 inhibitors used to treat T2D suppress *MTOR* and reduce pneumonia risk[38,39].

To test the potential causative effect of *MTOR* expression on T2D risk, we performed a summary data-based Mendelian randomisation (SMR) analysis[40]. We focused on the genes within a 2 Mb window around the GWAS risk loci of the *MTOR* locus and observed significant effects on increased T2D risk through higher expression of *MTOR* in activated T cells ($p_{SMR} = 1.17e{-}06$ and $1.92e{-}6$ in CD4+ and CD8+ T cells, respectively; Fig. 2f, g and Supplementary Table 4; see "Methods"). In line with the eQTL associations (Fig. 1f), we observed inverted

effects in neutrophils and adipose tissue (Fig. 2h and Supplementary Fig. 3c, d), both of which are known to interact with T cells in disease settings[15,41,42].

Given the disease and immunological associations, we hypothesised that this *MTOR* variant would have been under selection pressure. We observed higher frequency of the G-allele in Africans, compared to other populations, with allele frequencies of 0.91, 0.31, 0.21, 0.30, and 0.38 in Africans, Americans, East Asians, Europeans, and South Asians respectively (Supplementary Fig. 3e), as well as high genetic divergence (measured by fixation index, $F_{ST}$[43]) (Supplementary Fig. 3f, g). This would be consistent with potential functional consequences associated with survival fitness and disease-modulating allele frequency following migrations and genetic adaptations within human populations[44].

## Activation of sepsis neutrophils by T cells is dampened by mTOR inhibition and hypoxia

Our findings of a protective G-allele effect in patients with relatively reduced *MTOR* expression in activated T cells, and with higher expression in neutrophils, led us to hypothesise a potential role for T cells in driving sepsis-associated neutrophil phenotypes via the mTOR pathway (Fig. 2i). To test this hypothesis, we established an ex vivo co-culture model of neutrophils and T cells in which we could compare effects of co-culture with either neutrophils from sepsis patients or healthy donors, with primary T cells that were resting or activated (Fig. 3a and Supplementary Fig. 4a). We assayed three neutrophil surface markers, CD64, CD123, and PD-L1, previously associated with sepsis severity[45,46], as indicators of neutrophil activity. All

three markers exhibited upregulation in neutrophils isolated from sepsis patients compared to healthy controls, at both the protein (Fig. 3b) and mRNA levels (Supplementary Fig. 4e–g). We found that co-culture with activated T cells, relative to resting T cells, resulted in a further increase in marker expression on sepsis neutrophils (Fig. 3b, c and Supplementary Fig. 4b). This upregulation was associated with enhanced NETosis (Supplementary Fig. 5a). Consistent with the observed eQTL association, both CD4+ and CD8+ T cells demonstrated similar effects on neutrophil markers (Supplementary Fig. 4c, d). However, the immortalised Jurkat T cell line did not affect these markers on sepsis neutrophils compared to primary T cells (Supplementary Fig. 4c, d), potentially due to altered T cell function in the Jurkat line. Interestingly, co-culture with activated T cells did not significantly affect CD123 and CD64 expression on neutrophils from healthy donors or convalescent patients compared to resting T cells (Fig. 3c), suggesting a specific priming of sepsis neutrophils for hyperactivation by activated T cells.

T cell activation is known to be heavily reliant on mTOR and HIF-1α (hypoxia-inducible factor 1α)[26], and mTOR inhibition and hypoxia attenuate T cell activity and cytokine release[47,48]. Consistent with these previous findings, in co-culture assays in the presence of rapamycin, a specific mTOR inhibitor, under hypoxia condition, or with rapamycin pre-treated T cells, we observed a decreased effect on sepsis neutrophils (Fig. 3d, e and Supplementary Fig. 5b). This effect was specific to T cell-neutrophil interactions, as rapamycin treatment did not affect sepsis neutrophils cultured alone (Supplementary Fig. 5c). Additionally, this dose-dependent effect of rapamycin did not affect neutrophil viability (Supplementary Fig. 5e, f). Together, these results highlight the critical role of mTOR activity in T cell activation and the consequent deleterious effects on neutrophils in the context of sepsis.

## Reciprocal interactions between activated T cells and sepsis neutrophils

Previous studies have revealed that neutrophils regulate adaptive immune responses either by releasing secreted peptides and cytokines or by direct cell-surface interactions[15,16], and neutrophils from SRS1 sepsis patients exhibit a stronger inhibitory effect on T cell proliferation and activation relative to non-SRS1 patients[8]. In our co-culture experiments, PD-1 and CD69 expression was significantly suppressed on T cells in the presence of sepsis neutrophils, an effect not observed when using neutrophil-conditioned media alone (Fig. 4a), suggesting direct cell-surface contact is crucial. Conversely, CD123 and CD64 expression was markedly upregulated on sepsis neutrophils when co-cultured with activated T cells or exposed to conditioned T cell media (Fig. 4b). This upregulation was also observed in a sepsis whole blood culture model (Supplementary Fig. 6a, b), where T cell activation enhanced neutrophil markers (Supplementary Fig. 6c, d), and the increase in immunosuppressive neutrophils subsequently inhibited T cell activation (48 h vs. 24 h; Supplementary Fig. 6e). Together, these results indicate a negative feedback loop between T cell activation and neutrophils in sepsis, mediated by both cytokine release and direct cell-surface interactions (Fig. 4c). Disruption of this loop, due to excessive mTOR signalling in T cells, may contribute to poor clinical outcomes.

To prioritise the potential cytokines mediating the cell-cell interactions, we generated RNA-seq datasets in primary CD4+ and CD8+ T cells following activation with anti-CD3/CD28 beads. As expected, TCR stimulation triggered substantial transcriptomic reprogramming (Fig. 4d and Supplementary Table 5), with 68 and 59 differentially expressed (DE) genes encoding annotated cytokines identified in CD4+ and CD8+ cells, respectively. Overall, 81.4% (48 out of 59) of DE cytokines in CD8+ T cells overlapped with those in CD4+ cells (Fig. 4e) and displayed a strong correlation (Pearson's r = 0.84; Fig. 4f). Interestingly, 11 of the 48 commonly DE cytokines in both T cell subsets were also rapamycin-sensitive (Fig. 4g), including

IL-3 (ligand for its receptor CD123) and IFN-γ, a cytokine known to augment NETosis via ligand-receptor interaction, contributing to acute lung injury during sepsis[17]. Additionally, the transcription of the sepsis surface markers was also upregulated in response to T cell cytokines (Supplementary Fig. 5d), suggesting a more intricate regulatory network.

## Regulatory mechanism underlying the *MTOR* eQTL in activated T cells

We proceeded to investigate the regulatory genomic landscape of the *MTOR* genetic association in order to prioritise and further characterise the *MTOR* eQTL. Studies have shown that causal eQTLs and GWAS risk SNPs are more likely to be found in open chromatin[23,29]. We investigated context specific epigenomic datasets and found that the *MTOR* eQTL variant rs4845987 is located at the centre of an ATAC-seq peak in intron 8 of *MTOR*, a peak identified exclusively in resting T cells (both CD4+ and CD8+), but not in other immune cells (Fig. 5a, Supplementary Fig. 7a and Supplementary Table 6). This ATAC peak was completely silenced in activated T cells (Fig. 5a, b), coinciding with a specific reduction in *MTOR* expression, while the expression of other nearby genes remained unaffected (Fig. 5c, d and Supplementary Fig. 8b). We also observed a similar pattern of the ATAC peak at this location being present in central memory T cells, but not in naive cells, following TCR activation and CAR T cell production (Supplementary Fig. 7b, c). Upon stimulation, this locus exhibited coordinated changes in epigenetic features, marked by the absence of enhancer markers H3K27ac and H3K4me1 and presence of hydroxymethylation in activated T cells (Fig. 5e and Supplementary Fig. 8c). The importance of this specific genomic region was further supported by proximal CTCF-binding and its interaction with the *MTOR* promoter (Fig. 5e and Supplementary Fig. 8a).

We hypothesised that the enhancer plays a critical role in maintaining *MTOR* expression and cell survival in resting memory T cells while masking the eQTL effect within the enhancer itself, and that in activated T cells, the eQTL-associated genetic variant may exert its effect through hydroxymethylation, leading to an allelic effect observed in bulk T cells (Fig. 1f) and single-cell T cell subsets (Supplementary Fig. 7d). To experimentally confirm this, we manipulated hydroxymethylation using vitamin C (VC), a known activator of TET enzymes that catalyse demethylation and 5-hydroxymethylcytosine (5hmC)[49]. As predicted, we observed increased *MTOR* expression upon VC in activated T cells with the C/C genotype compared to G allele carriers (Fig. 5f), and knockdown of *MTOR* by around 40% using siRNAs (Supplementary Fig. 9a) led to a significant reduction in IFN-γ release from activated T cells carrying the C allele, both at baseline and following VC treatment (Fig. 5g).

We next utilised a CRISPR/dCas9-based epigenetic activation system delivered via lentivirus to manipulate this locus and demonstrate direct evidence for enhancer function (Fig. 5h; see Methods). We observed a significant upregulation of *MTOR* expression in activated T cells with sgRNA targeting the variant-containing regions (sgRNA-e) compared with a non-targeting control sgRNAs, or sgRNAs targeting the flanking regions (Fig. 5i). To directly test the allelic effect of the prioritised eQTL SNP rs4845987 on *MTOR* expression, we utilised a base-editing approach to introduce a C-to-T substitution at the endogenous locus[50]. We co-delivered SpG-TadCBE6b mRNA and sgRNAs precisely targeting the variant in activated T cells with C/C genotype (Fig. 5j; see "Methods"). Efficient C-to-T conversions were observed at both the SNP site (Fig. 5k) and a control site located -100 bp upstream (Supplementary Fig. 9b). We found that editing at the SNP resulted in a ~20% reduction in *MTOR* expression compared with editing of the upstream control region or non-edited cells. A minor bystander C-to-T conversion was detected 5 bp downstream of the SNP, but this cytosine lacks any known regulatory or functional association (Supplementary Fig. 9c).

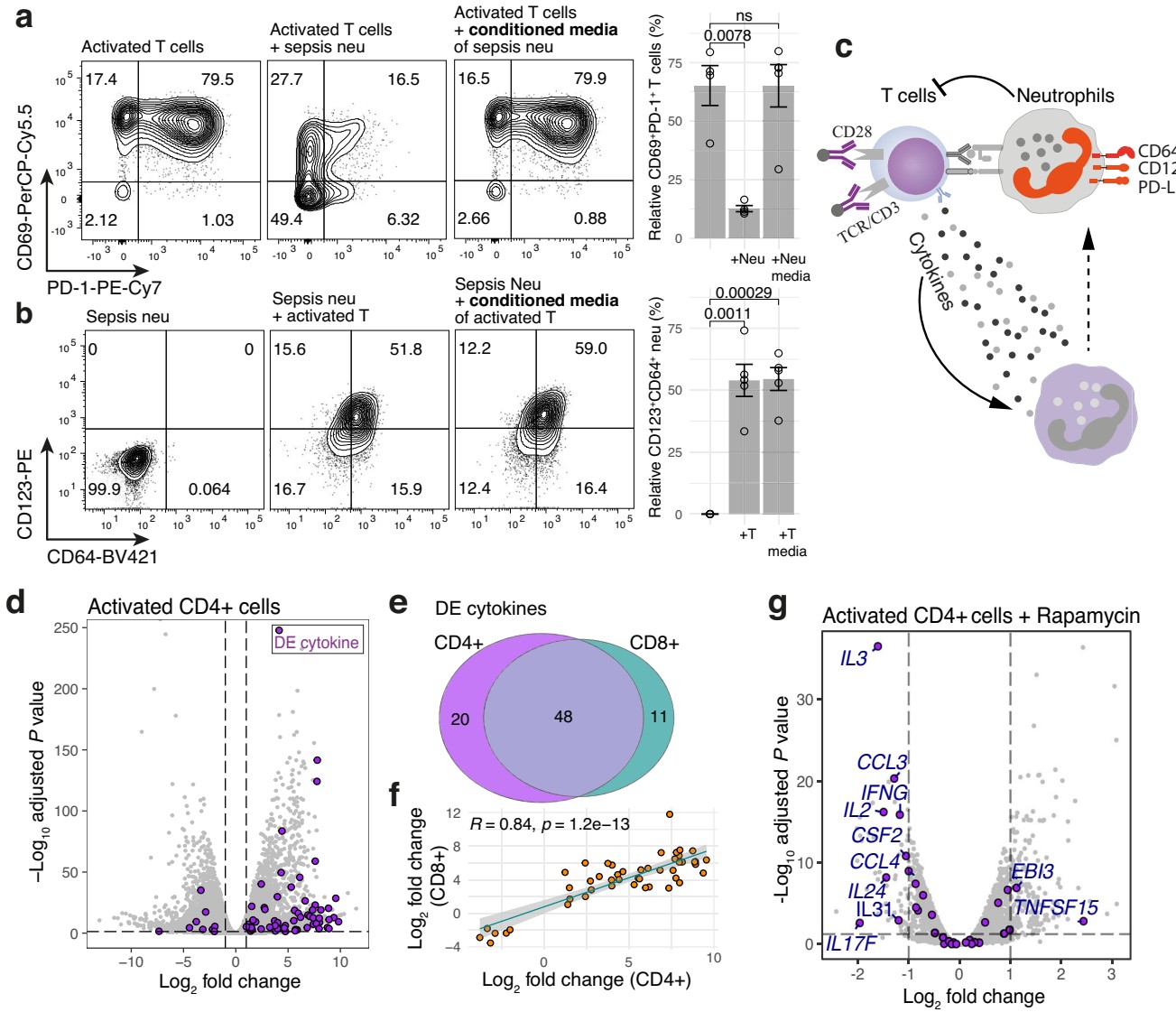

**Fig. 4 | Reciprocal interactions between activated T cells and sepsis neutrophils. a** Flow cytometry panels showing the inhibition of T cell activation (CD69 and PD-1 expression) in the presence of sepsis neutrophils, but not with conditioned neutrophil media cultured for 2 days. Bar plot (right panel) with the percentage of activated T cells co-cultured with or w/o sepsis neutrophils or conditioned neutrophil media ($n = 5$ from 5 patients). **b** Flow cytometry results showing the activation of sepsis neutrophils (CD123 and CD64 expression) by activated CD8+ T cells. Bar plot (right panel) for the percentage of activated neutrophils co-cultured with or w/o activated T cells or conditioned T cell media. Each dot represents an independent replicate using neutrophils from individual patients ($n = 5$). $p$ value was calculated by two-tailed $t$-test. Error bars represent SEM of the replicates. **c** Interplay between sepsis neutrophils and allogeneic T cells upon TCR

activation. **d** Volcano plot showing differentially expressed (DE) genes upon TCR activation in CD4+ T cells derived from three healthy volunteers (left panel). DE genes (FDR < 0.05, Fold change >2) encoding cytokines annotated by UniProtKB ($n = 189$: searched using Cytokine (KW-0202) for ones reviewed by Swiss-Prot) were highlighted in purple. **e** Venn diagram showing the overlap of DE cytokines between CD4+ and CD8+ T cells. **f** Correlation of log2 fold change of DE cytokines identified in CD4+ and CD8+ T cells upon activation. Shaded bands indicate the 95% confidence interval (CI) of the fitted linear regression. Pearson's r and $p$ values are shown. **g** Volcano plot showing differentially expressed genes upon rapamycin (100 nM) treatment in activated T cells with anti-CD3/CD28 beads ($n = 3$). RNA-seq raw data was downloaded from GSE129829[85]. Source data for **a**, **b** are provided as a Source Data file. See also Supplementary Fig. 6.

Together, these results demonstrate a unique genetic and epigenetic mechanism that ensures precise control of *MTOR* expression, which is critical for proper T cell function in immune responses (Fig. 6).

## Discussion

mTOR is a critical signalling hub that integrates immune receptor and metabolic signals to determine the fate of T cells[26]. In this study, we utilised epigenetic profiling to identify a regulatory element that controls *MTOR* transcription, distinguishing between resting and activated T cells. This enhancer is essential for memory T cell survival and functions as a negative feedback loop to suppress mTOR activation upon TCR stimulation. Further studies are needed to elucidate key

extrinsic modulators by examining epigenetic remodelling of cytokine loci, with a focus on the mTOR pathway and those affecting sepsis-associated neutrophil dysfunction. The deleterious effects of cytokine release during sepsis are supported by recent observations of sustained cytokine signalling upregulation and the potential harmful effects of interferon gamma-1b therapy in pneumonia cases[51,52]. Understanding these modulators could also provide insights into whether these factors impact the efficacy of CAR-T cell therapy, which is often complicated by severe infection[53]. Pretreatment of CAR-T cells with rapamycin has shown promise in promoting bone marrow infiltration and enhancing therapeutic efficacy in acute myeloid leukaemia[54]. Our data demonstrate that selective modulation of

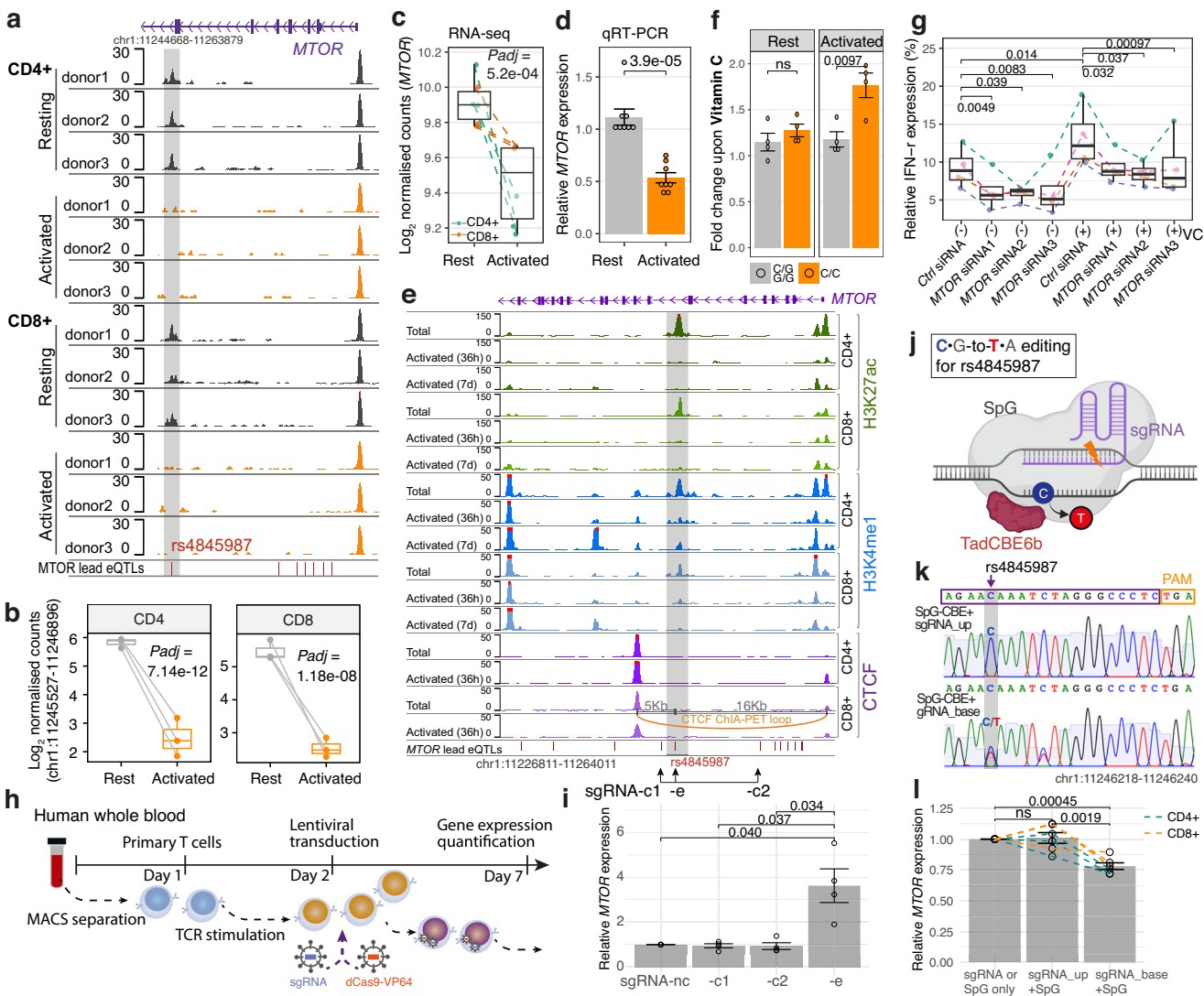

**Fig. 5 | Regulatory mechanisms underlying the *MTOR* eQTL in activated T cells.**
**a** ATAC-seq showing an *MTOR* eQTL resides in a differential open chromatin (in grey) of T cells following anti-CD3/CD28 stimulation. Quantifications of the ATAC peak (**b**) (highlighted in **a**) and **c** the expression of its proximal gene *MTOR* in activated T cells relative to resting T cells (*n* = 6 from 3 healthy volunteers). Adjusted *p* values were calculated using *glm.nb* and *Wald's test* as implemented in *DESeq2*. **d** *MTOR* transcripts measured using qRT-PCR normalised to β-Actin in activated CD4+ T cells (anti-CD3/CD28 Dynabeads for 3 days) relative to resting T cells ($2^{-\Delta\Delta Ct}$) extracted from healthy donors (*n* = 8). Error bars represent SEM of the replicates. **e** Epigenetic status surrounding the *MTOR* eQTL locus in T cells following anti-CD3/CD28 activation. ChIP-seq for histone modifications and CTCF binding were downloaded from the Encyclopedia of DNA Elements (ENCODE) project[86] (see Supplementary Table 7). *MTOR* lead eQTLs are represented by red bars (see also Supplementary Figs. 1a, b and 8a for the full locus). **f** *MTOR* transcripts in CD4+ T cells treated with vitamin C (200 uM for 48 h) relative to the vehicle control. Each dot represents an individual healthy donor (*n* = 8). Error bars represent SEM of the replicates. *p* value was calculated by two-tailed *t*-test. **g** Quantification of relative IFN-γ release in T cells carrying the C allele. T cells were pre-activated for 2 days and electroporated with three siRNAs targeting *MTOR*. After 6 days, cells were harvested with or without 2 days of VC treatment. Brefeldin

A was added before harvest to block cytokine secretion, followed by intracellular IFN-γ staining and flow cytometry (see "Methods"). Each dot represents an independent replicate (*n* = 4 from 2 donors). *p* value was calculated by two-tailed paired *t*-test. **h** Schematic of CRISPR/dCas9-mediated epigenetic gene activation and the delivery strategy in primary T cells. **i** *MTOR* transcripts in CRISPRa (dCas9-VP64) edited CD4+ T cells with an sgRNAs targeting the variant-containing region (sgRNA-e; highlighted in **e**), or sgRNAs targeting the surrounding regions (sgRNA-c1 and -c2) compared to a non-target sgRNA control (sgRNA-nc) (see Supplementary Table 8 for sgRNA sequences). Error bars represent SEM of 4 independent replicates from 2 healthy donors. **j** Schematic of SpG-TadCBE6b mediated base editing. Created in BioRender. Zhang, P. (2026) https://BioRender.com/1j3qxv0. **k** Sanger sequencing chromatogram showing base edits in T cells with the C/C genotype delivered with SpG-TadCBE6b mRNA and sgRNAs targeting SNP rs4845987 (sgRNA-base). Protospacer-adjacent motif (PAM) is highlighted in orange. **l** *MTOR* transcripts in T cells edited with SpG-TadCBE6b and sgRNA-base compared with an sgRNA targeting a site 100 bp upstream (sgRNA-up), or with cells electroporated with either SpG-TadCBE6b or sgRNA-base alone (see "Methods"). Error bars represent SEM of 6 independent replicates from CD4+ or CD8+ T cells derived from 3 healthy donors. *p* value was calculated by two-tailed *t*-test. Source data for **d**, **f**, **g**, **i**, **l** are provided as a Source Data file. See also Supplementary Figs. 7–9.

excessive mTOR signalling in T cells by rapamycin alleviates neutrophil hyperactivation, potentially restoring immune balance without broad immunosuppression and highlighting T cell-neutrophil crosstalk as a therapeutic target in sepsis due to pneumonia.

We observed the genetic association of the *MTOR* SNP with survival only in patients with pneumonia, suggesting that tissue-resident

immune interactions within the lung may play a key role in mediating this organ-specific genetic effect. Dysregulated neutrophil activation and crosstalk with adaptive immune cells could amplify local inflammatory responses, thereby modulating the impact of mTOR-dependent pathways on sepsis outcomes during pneumonic infection. Previous work indicates that immune suppressive monocytes

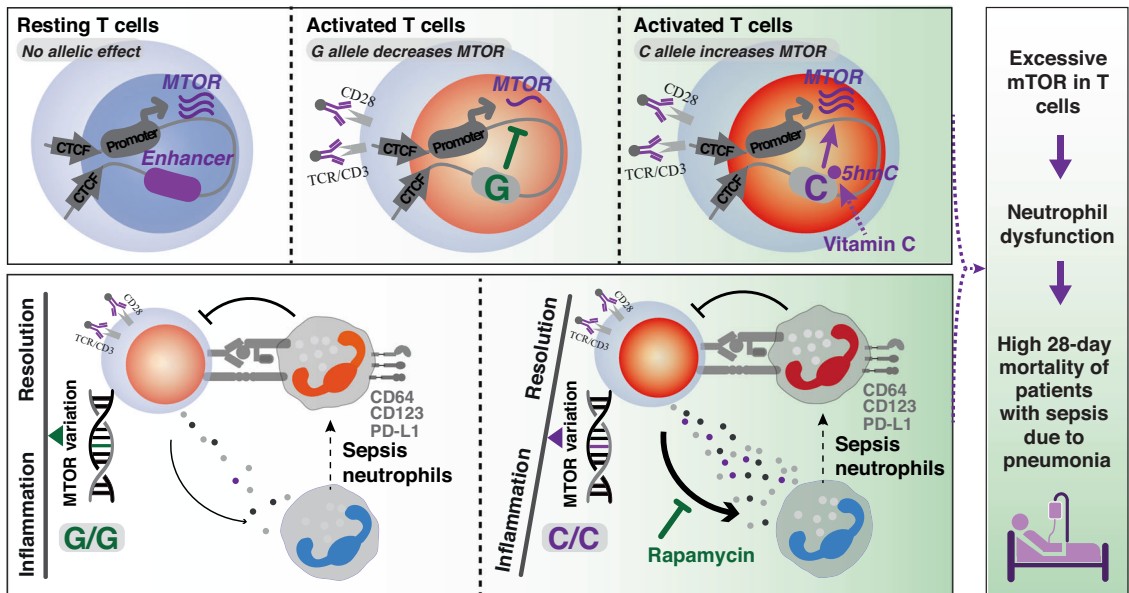

**Fig. 6 | Potential model for genetic regulation of *MTOR* in T cells and T cell-neutrophil crosstalk during sepsis.** An enhancer element regulates *MTOR* transcription in T cells, maintaining baseline mTOR activity in resting T cells while masking the *MTOR* eQTL effect. Upon T-cell receptor activation, mTOR is highly upregulated through post-translational modifications, with silencing of this enhancer providing a negative feedback loop to prevent excessive mTOR activity at the transcription level. In activated T cells, hydroxymethylation exposes the eQTL effect of this common genetic variant (minor allele frequency = 0.39), which can be modulated by vitamin C. The C allele amplifies mTOR signalling and T cell activity during sepsis, ultimately leading to persistent neutrophil hyperactivation, NETosis, and increased mortality in patients with sepsis due to pneumonia.

show prognostic significance in sepsis[1,2] and that prolonged endotoxin exposure induces tolerance in monocytes[18,55], a setting in which the *MTOR* eQTL effect was also observed. This further underscores the importance of considering cell-type-specific and subset-specific regulatory mechanisms. We demonstrated allele-specific epigenomic regulation in T cells, but the molecular mechanisms underlying the inverted effect in neutrophils and monocytes remain unclear. Future studies incorporating single-cell epigenomic, cytokine profiling and humanised mouse models will be critical to dissect the functional heterogeneity of immune cell populations and to define the cellular and molecular mechanisms underlying *MTOR*-dependent genetic regulation and complex immune interactions.

There are a number of limitations to our study. TadCBEs preferentially edit cytosines within positions 4-8 (5′ to 3′) of the protospacer from the PAM-distal end of the sgRNA[50]. We observed a bystander C-to-T conversion 5 bp downstream of the target SNP, outside this canonical window, in primary T cells. As this cytosine has no known regulatory link to *MTOR* expression, it is unlikely to underlie the observed effect. Nonetheless, this represents a technical limitation. Future optimisation could utilise narrower editors such as TadCBEd though with reduced editing efficiency[50], or PAM-relaxed Cas9 variants such as SpRY[56] to reposition the protospacer and avoid nearby cytosines. In addition, further whole-genome sequencing will be required to assess potential off-target edits introduced by the new SpG-TadCBE6b fusion in T cells. A further limitation is that our analysis was performed at the bulk level, which may obscure heterogeneity among cell types or subsets. Single-cell approaches in future studies could help delineate cell type-specific dynamics during T cell differentiation and acute infection. For the co-culture experiments examining reciprocal Neu-T crosstalk, the 2-day culture interval was selected because it maintains high neutrophil viability while still allowing robust cytokine production from anti-CD3/CD28-stimulated primary T cells. Progressive neutrophil apoptosis and loss of function with time is an important potential confounder in such experiments.

Chronic inflammation and metabolic dysfunction in adipose tissue are key drivers of insulin resistance and T2D[57]. Both CD4+ and CD8+ T cells infiltrate adipose tissue, with increased interferon-γ-expressing T cells promoting inflammation[58,59]. The adipocyte-secreted hormone leptin directly regulates T cell proliferation, glycolytic metabolism and cytokine production[41,42]. Individuals with T2D have an increased risk of developing infections including sepsis and COVID-19[60]. However, the overall evidence linking T2D to sepsis outcomes remains unclear but appears more robust for pneumonia[61,62]. Interestingly, sodium-glucose cotransporter-2 inhibitors used to treat T2D suppress *MTOR*[39] and reduce pneumonia risk[38]. Our study demonstrated colocalisation of a genetic variant between adipose tissue, T cells for *MTOR* expression and T2D GWAS loci. This tissue-specific variant had opposite allelic effects on *MTOR* expression, with the G allele decreasing *MTOR* expression in activated T cells while increasing it in adipose tissue. These results suggest a potential role for mTOR signalling in the adipose-T cell axis with implications for the interplay between sepsis and T2D.

Vitamin C (VC) supplementation has been proposed as a potential treatment for sepsis due to its anti-inflammatory and antioxidant properties. However, clinical trials have shown mixed results[63–66], potentially due to different patient severities at enrolment. The LOVIT trial[63] reported detrimental effects of VC treatment in sepsis, particularly among patients without septic shock, consistent with our finding that the MTOR risk variant was associated with mortality predominantly in immunocompetent and non-SRS1 individuals. We further showed that VC increases *MTOR* expression and *MTOR*-dependent interferon-γ release in activated T cells from risk C-allele carriers, providing insights into the potential mechanism of action of harmful VC therapy in this subgroup. These observations point toward genotype- and biomarker-guided patient stratification as a potential route to clinical translation, as rs4845987 genotyping could be rapidly implemented using PCR-based or targeted sequencing assays on admission blood samples to identify C/C carriers. In contrast to VC, which activates TET enzymes[49], TET inhibition could potentially reduce T cell-mediated cytokine release, alleviate neutrophil dysfunction, and improve patient survival. Indeed, treatment with TET inhibitors reduces 5hmC levels and attenuates LPS-induced pulmonary

oedema and lung injury in a sepsis mouse model[67]. However, the specific role of TET inhibition in modulating T cell-neutrophil crosstalk during human sepsis remains unclear. Exploring genotype-dependent links between 5hmC and VC response through allele-specific epigenomic and temporal TET activity analyses could further delineate this mechanism. Ultimately, a randomised clinical trial will be essential to confirm the in vivo relevance and translational success of rs4845987 genotyping or mTOR-related biomarkers.

## Methods

### eQTL interaction and colocalisation analysis

eQTL interaction was determined using a linear mixed model as implemented in R package lmerTest[68]: p ~ g + i + g:i + (1|donor), where p is the gene expression corrected for population structure and PEER factors as described in ref. 25, g is the genotype vector, i is the term for SRS endotypes or log2 transformed NLR, g:i is the interaction between genotype and either SRS or log2 transformed NLR, and (1|donor) is a random effect accounting for variability between donors. Gene expression and genotype data were obtained from 823 RNA-seq samples derived from 638 sepsis patients, some of whom contributed multiple longitudinal samples, as described by Burnham et al.[25]. SRS and SRSq scores were estimated using a 7-gene signature implemented through SepstratifieR[7]. Among the 638 GAinS patients, 36.7% and 44.4% were classified as SRS1 using the SRS-latest and SRS1-ever assignments respectively.

Colocalisation analysis was performed in a ± 200-bp window surrounding the *MTOR* lead eQTL, using R package coloc[69] with default settings. eQTL summary statistics from healthy tissue and cell types were retrieved from eQTL Catalogue[29]. Type 2 diabetes GWAS summary statistics were downloaded from https://ftp.ncbi.nlm.nih.gov/dbgap/studies/phs001672/analyses/[36], and https://www.diagram-consortium.org/downloads.html[37].

### Cell type deconvolution

Aligned bam files derived from sepsis leucocytes were obtained from the GAinS study. RNA-seq count matrix was generated using feature-Counts and normalised by DESeq2. The CIBERSORTx[70] absolute scores for each cell type in bulk RNA-seq samples were estimated using the bulk count matrix and the signature matrix of 23 distinct cell subsets excluding apoptosing cells derived from sepsis single cell RNA-seq[8] with S-mode batch correction and 100 permutations. Sample-level absolute scores across subsets were compared between SRS1 and non-SRS1 groups using a linear mixed model accounting for donor variability (Supplementary Fig. 10).

### Summary data-based Mendelian randomisation

T2D GWAS in Europeans and eQTL summary data were obtained from Suzuki et al.[37] and eQTL Catalogue[29], respectively. We performed the SMR analysis[40] using eQTLs as instrumental variables to identify genes whose expression is associated with T2D risk due to pleiotropy and/or causality. Genes were included in the analysis if they had at least one cis-eQTL ($p < 5e^{-8}$) in either activated CD4+, CD8+ T cells[20], neutrophils[71], or adipose/fat tissue[72,73] within a 2 Mbp window around GWAS loci, following the default settings of the SMR tool (v1.3.1). The HEIDI (heterogeneity in dependent instruments) test was applied to differentiate functional associations from linkage effects. LD correlation between SNPs was estimated using 1000 Genomes Project data for Europeans.

### UK Biobank (UKB) data curation and analysis

For UK Biobank data, collected following established recruitment and ethical approval procedures[74], half a million men and women aged 40–69 years attended one of 22 UKB assessment centres located throughout England, Scotland and Wales between 2006 and 2010. All participants completed a touchscreen questionnaire, verbal interview

and had a range of physical measurements and blood, urine and saliva samples taken for long-term storage. The UK Biobank genetic data contains genotypes for 488,377 participants[75]. Withdrawn participants were removed from the available UKB dataset and only individuals who received a diagnosis of bacterial pneumonia based on International Classification of Diseases 10th Revision (ICD10) J15 in any instance of the UKB-specified Field 41270 were included in the analysis (n = 1461). The corresponding date of diagnosis was extracted from the matching instance in Field 41280. For those with multiple diagnoses the sole or earliest date of diagnosis was used as the only reference point for impatient start date. Participant death was determined via the presence of a date in instance 0 of Field 40000. Individuals who died within 28 days of their earliest ever recorded diagnosis of the relevant ICD10 code were classified as a case with the remainder as controls. The time in days between the earliest date of diagnosis and date of death was used as the time to event metric. Age in years was calculated by subtracting the estimated date of birth (using 'year-of_birth_f34_0_0', 'month_of_birth_f52_0_0', and day of birth manually assigned as the 15th to avoid regression to the mean) from the date of the relevant ICD10 diagnosis and dividing by 52.143. Self-reported sex in instance 0 of field 31 was used to define males and females. Patients with a genetically determined ancestry of White British Subset (WBS) were included (n = 1125), and the first seven principal components were used for survival analysis as described above. Immunosuppressed status (IS) was defined based on ICD-10 digital codes (Field 41270)[76], covering chronic viral hepatitis, Human Immunodeficiency virus (HIV) or Human T-cell lymphotropic virus (HTLV) infection, organ transplantation, medication, autoimmunity, blood cancer, and congenital disease. T2D status is allocated via broad top level ICD10 (E11), using the same data fields available for IS using ICD10 digital codes from diagnosis (Field 41270) and date fields 41280. The cancer category is any type of cancer diagnosed before the sepsis/case definition according to the UK cancer register, using diagnosis fields 40006 and date fields 40005.

### SNP genotyping

We genotyped the SNP rs4845987 for the Sepsis Immunomics (SI) cohort using samples derived from CAP patients with hospitalisation dates up to 04-12-2024. The inclusion and exclusion criteria for this study were detailed above. Briefly, genomic DNA was extracted from buffy coat samples (stored at −80 °C) using the Monarch Genomic DNA Purification Kit (NEB #T3010) following the manufacturer's instruction. gDNA was then quantified using the Qubit dsDNA HS Assay Kit (ThermoFisher). Sample genotypes were determined using Taq-Man Genotyping Assays with the Universal PCR Master Mix and a predesigned probe for rs4845987 (C_2524855_10, ThermoFisher) on a CFX-96 C1000 platform (Bio-Rad). For healthy volunteers, purified cells were used for gDNA extraction.

### Survival analysis

Patients with sepsis were recruited from the Genomic Advances in Sepsis (GAinS) and Sepsis Immunomics (SI) studies, following established inclusion and exclusion criteria[8,77]. The GAinS cohort included 1168 patients with genotype data available (54% men; aged 18–92 years), predominantly of European ancestry (96%). The SI-CAP cohort used in this study comprised 102 European patients (52% male; aged 19–98 years) who were genotyped for the *MTOR* SNP rs4845987. Blood samples were obtained on Day 1, 3 and 5 of hospital admission, with follow-up samples taken an average of 3 months after hospital discharge. CAP was diagnosed as a febrile illness with cough, sputum production, breathlessness, leukocytosis and radiological evidence of pneumonia, acquired in the community or within 2 days of hospital admission. To assess the association between genetic variants and 28-day mortality, Cox proportional-hazards model and logistic regression adjusting for age, sex, and the first seven principal components were

used. Multivariable Cox regression was performed to evaluate the predictive ability of hypoxia-related factors (FiO2, PaO2:FiO2 ratio) for 28-day mortality while controlling for age (>65 vs. ≤65 years) and sex. All statistical analyses were conducted using R (v4.2.1).

## Isolation and culture of human neutrophils and T cells

Peripheral blood mononuclear cells (PBMCs) from healthy donor leucocyte cones were isolated by Ficoll-Paque (Sigma) density gradient centrifugation. CD4+ and CD8+ T cells were then separated from PBMCs by positive selection with magnetic MicroBeads (Miltenyi Biotec) following the manufacturer's instructions. Isolated T cells were frozen in FBS (Sigma, #F7524) with 10% DMSO at $2 \times 10^7$ cells/ml and stored in liquid nitrogen. Total T cells were maintained in RPMI-1640 (Gibco) medium supplemented with 5% FBS (Sigma, #F7524), 100 mM L-glutamine (Sigma), 1x penicillin-streptomycin (Sigma) and 500 IU/ml human recombinant IL-2 (Biolegend). T cells were activated using anti-CD3/CD28 Dynabeads (ThermoFisher, #11131D) at a 1:1 cell:bead ratio at $2 \times 10^6$ cells/ml.

Neutrophils were extracted from 2–10 ml whole blood from sepsis patients or healthy donors using EasySep HLA Chimerism Whole Blood CD66b positive selection kit (STEMCELL) as per manufacturer's instructions. Neutrophils were maintained in RPMI-1640 (Gibco) medium supplemented with 10% FBS, 100 mM L-glutamine and 1x penicillin-streptomycin. For co-culture, cryopreserved CD4$^+$ T cells were thawed and cultured with CD66b$^+$ neutrophils in 24-well plates at a 1:2 T cells to neutrophils ratio in the presence or absence of anti-CD3/CD28 beads for 48 h. Co-culture was performed for 2 days unless otherwise indicated. Rapamycin (#A8167-APE) was obtained from Stratech Scientific. L-ascorbate (Vitamin C) was from Merck (#11140).

## Flow cytometry

Cells were harvested by centrifugation at $300 \times g$ for 3 min, followed by live/dead cell staining (LIVE/DEAD™ Fixable Green Dead Cell Stain Kit; ThermoFisher). Cells were stained using surface makers CD3-APC (Biolegend, #300412), PD-1-PE-Cy7 (Biolegend, #367414), CD69-PerCP-Cy5.5 (Biolegend, #310926), CD66b-AF700 (Biolegend, #305114), PD-L1-BV605 (Biolegend, #329724), CD123-PE (BD Biosciences, #554529) and CD64-BV421 (Biolegend, #305020) for 45 min at room temperature, and washed with 0.2% bovine serum albumin (BSA) in phosphate-buffered saline (PBS). Samples were then acquired on a LSRFortessa X-20 (BD Biosciences) flow cytometer and analysed using FlowJo software (v10.10). Whole blood samples from sepsis patients were collected in BD Vacutainer EDTA tubes and cultured in T cell media at a 1:4 ratio, with or without anti-CD3/CD38 Dynabeads at a 1:10 v/v ratio, for 48 h. Cultures were then treated twice with RBC lysis buffer prior to antibody staining and flow cytometry. For IFN-γ quantification, cells were incubated with BD GolgiPlug (brefeldin A) for 4 h, stained for live/dead dye and surface markers, then fixed and permeabilised using the BD Cytofix/Cytoperm kit, followed by overnight staining with IFN-γ-PE (BioLegend, #506507).

## qRT-PCR and RNA-seq

Total RNA was extracted from lysed cells using the Monarch Total RNA Miniprep Kit (NEB #T2010). cDNA synthesis was subsequently performed with LunaScript RT SuperMix Kit (NEB #E3010). Gene expression levels were quantified by qRT-PCR using SYBR Green Real-Time PCR Master Mix (Qiagen) on a CFX-96 C1000 platform (Bio-Rad), with β-actin as the normalisation control (see Supplementary Table 8 for primer sequences).

One μg RNA was used for library preparation using the NEBNext Ultra II RNA Library Prep Kit for Illumina (#E7770S) following the manufacturer's protocol. Poly(A) mRNA enrichment was performed using the NEBNext Poly(A) mRNA Magnetic Isolation Module (#E7490). Library quality control, including size distribution and quantification, was assessed using TapeStation 4200 (Agilent) with

High Sensitivity D1000 reagents and Qubit HS DNA kit (Thermo Fisher), respectively. Final library molarity was determined using the KAPA Library Quantification Kit (Roche). Libraries were sequenced on the Illumina NextSeq 500 platform using a 150-cycle High Output Kit v2.5.

## RNA-seq analysis

Raw RNA-seq reads were trimmed using Trim Galore (v0.6.2) and aligned to the human genome (hg38) using HISAT2 (v2.1.0). Transcript quantification was performed using featureCounts (v1.6.2) with GEN-CODE v31 annotations. The bigwig files normalised by RPKM (Reads Per Kilobase per Million mapped reads) were generated using the bamCoverage function of deepTools (version 3.3.1). Differential gene expression analysis was conducted using DESeq2 (v1.36.1) on raw read counts.

## Omni-ATAC-seq and analysis

A total of 50,000 cells were prepared by centrifugation and resuspended in 50 μl of lysis buffer (10 mM Tris-HCL pH 7.4, 10 mM NaCl, 3 mM MgCl2, 0.01% Digitonin, 0.1% Tween-20 and 0.1% Igepal CA-630) for nuclear isolation. Following transposition and DNA purification, library preparation was performed using the Omni-ATAC protocols[78,79]. DNA libraries were purified using a MinElute PCR Purification Kit (Qiagen) and AMPure XP Magnetic Beads (Beckman Coulter), quantified by Qubit assays (Thermo Fisher Scientific), and quality assessed with an Agilent TapeStation. Final libraries were sequenced on the Illumina NextSeq 500 platform. Sequencing reads for ATAC-seq were aligned to the human genome (hg38) using Bowtie2 (v2.2.5). data were filtered for quality control using Picard (v2.0.1) and Samtools (v1.9) before peak calling with MACS2 (v2.1.0)[23,79]. Differential peak analysis was performed using DESeq2, considering peaks present in at least 30% of samples. Potential batch effects and/or technical variation were assessed through principal component analysis and incorporated as covariates in the DESeq2 design formula.

## Public hMeDIP-seq analysis

Raw sequencing reads for hydroxymethylated DNA Immunoprecipitation Sequencing (hMeDIP) data were obtained from GSE74850[80], trimmed using Trim Galore (v0.6.2), and aligned to human genome (hg38) using the BWA-mem alignment algorithm (v0.7.12)[81]. The binary alignment and map (BAM) files were filtered to remove reads with a mapping quality score less than 10 and duplicate reads using SAMtools (v1.9) and Picard (v2.21.1). The normalised fold enrichment tracks over the corresponding input controls were generated by using the *callpeak* function with the *--SPMR* flag, then passing the bedgraph outputs into the *bdgcmp* function of MACS2 and the bedGraphToBigWig tool.

## Lentivirus production

Human embryonic kidney (HEK) 293FT cells (Sigma #12022001) were maintained in Opti-MEM I Reduced Serum Medium (OPTI-MEM) with GlutaMAX Supplement (ThermoFisher, #51985034) supplemented with 5% FBS (Sigma, #F7524), 1 mM sodium pyruvate (ThermoFisher), and 1× MEM nonessential amino acids (ThermoFisher) in T175 flasks. Cells were seeded per 150 mm dish (Corning, #430599) in 14 ml of medium overnight to achieve confluency about 90% at the time point of transfection. Cells were transfected with a plasmid mixture containing 5.7 μg psPAX2 (Addgene #12260), 3.2 μg pCMV-VSV-G (Addgene #8454), and either 4.6 μg of a sgRNA expression vector (Addgene #96923) or 7.0 μg of dCas9-VP64 (Addgene #180263) in equimolar ratios using jetPRIME reagent (Polyplus). After 6 h, the transfection medium was replaced with fresh medium supplemented with ViralBoost (Alstem Bio, #VB100). Lentiviral supernatant was harvested in 24 h post-transfection, filtered with a 0.45 μm membrane filter (Millipore), and concentrated using ultracentrifugation (Beckman Optima L-90K, SW32 Ti rotor) at 29,000 rpm for 2 h at 4 °C. The

pellet was resuspended in PBS with 1.5% BSA, aliquoted and stored at −80 °C. Lentivirus titre of dCas9-VP64 were determined by quantifying mCherry-positive cells via flow cytometry post-transduction. Viral volume that results in at least 50% transduction efficiency was used.

### CRISPR-dCas9 mediated epigenetic editing for primary T cells
We designed and selected top ranked single guide RNA (sgRNA) based on the scoring metrics using FlashFry[82]. For the human U6 promoter-based transcription, a guanine base was added to the 5′ of the sgRNA when the 20 bp guide sequence did not begin with G. The sgRNA sequences are listed in Supplementary Table 8. Primary human CD4+ T cells were activated using anti-CD3/CD28 Dynabeads (Thermo-Fisher). One million cells were transduced with sgRNA- and dCas9-VP64-containing lentiviruses in the presence of 8 μg/mL polybrene (Merck, #28728-55-4) for 24 h. Transduced cells were maintained in media and assayed in 6 or 7 days.

### Base editing using SpG-TadCBE6b
To generate a plasmid for in vitro transcription of a suitable base editor mRNA, the PEmax coding sequence was removed from pT7-PEmax (Addgene #178113) using SpRYgest[83] and the plasmid backbone was gel extracted. TadCBE6b and UGI (uracil glycosylase inhibitor) domains were amplified from SpCas9-CBE6b (Addgene #215820)[50], and the SpG Cas9 variant from pRDA_479 (Addgene #179099)[84]. These fragments were then cloned into the IVT backbone via isothermal assembly using the NEBuilder HiFi DNA Assembly Mastermix (NEB), according to manufacturer's instructions. SpG-TadCBE6b mRNA was synthesised from this plasmid using the HiScribe T7 mRNA Kit (NEB), replacing UTP with N1-methyl-pseudouridine (NEB #N0431).

Primary T cells were pre-activated with anti-CD3/CD28 beads for 2 days before electroporation. The synthesised SpG-TadCBE6b mRNA (2 μg) and sgRNA (150 pmol; Synthego) were then co-delivered using the P3 Primary Cell 4D-Nucleofector Kit (Lonza, #V4XP-3032). Cells were harvested 6 days post-electroporation for downstream analyses. sgRNA sequences for base editing, SpRYgest and oligonucleotide sequences for cloning and PCR primers for Sanger sequencing can be found in Supplementary Table 8.

### Statistics and reproducibility
Data are presented as bar plots showing the mean ± SEM or as box-and-whisker plots. Box plots show the median (centre line), interquartile range (box), and whiskers extending to 1.5× the interquartile range. Statistical analyses and data visualisation were performed using R (v4.2.1). Statistical tests used for each experiment are indicated in the figure legends and were two-sided. $p$ values are reported as unadjusted unless explicitly stated otherwise, in which case adjusted $p$ values ($p$adj) were calculated using false discovery rate (FDR) or Bonferroni correction as indicated. No statistical methods were used to predetermine sample size. There was no a priori allocation of participants to separate groups. Stratified analyses were performed based on the presence or absence of defined phenotypes. Transcriptional profiling, SRS assignment and genotyping were performed with blinding to clinical metadata.

### Ethics approval and consent to participate
Peripheral blood samples were obtained from healthy volunteers following informed consent (Oxfordshire Research Ethics Committee approval REC reference 06/Q1605/55); and from sepsis patients in the Sepsis Immunomics (SI) study (South Central Oxford REC C, reference:19/SC/0296) and UK GAinS (REC approvals 05/MRE00/38, 08/H0505/78, and 06/Q1605/55) with ethics approval granted nationally and locally, and informed consent obtained from all patients or their legal representative. UK Biobank has obtained ethics approval from the North West Multi-centre Research Ethics Committee (approval number: 11/NW/0382) and had obtained informed consent from all participants.

### Reporting summary
Further information on research design is available in the Nature Portfolio Reporting Summary linked to this article.

## Data availability
GAinS gene expression and genotyping data were deposited at the European Genome-phenome Archive (EGA), under accession numbers EGAD00001008730[7] and EGAD00001015369[25]. RNA-seq and ATAC-seq raw FASTQ files for CD4+ and CD8+ T cells are available under accession number EGAS50000000894. Processed data, including raw and normalised counts and bigWig files for genome-wide signal data, can be accessed on Zenodo (https://zenodo.org/uploads/14907264). The raw ATAC-seq data for primary immune cells were obtained from GSE172116 (macrophages), EGAS00001007362 (monocytes), GSE150018 (neutrophils), GSE118189 (NK and dendritic cells), and GSE168882 (CAR T cells). RNA-seq data in rapamycin-treated CD4+ T cells were obtained from GSE129829. MeDIP-Seq for 5hmC in CD4+ T cells was obtained from GSE74850. The processed histone modification and CTCF ChIP-seq results were downloaded from the ENCODE project (https://www.encodeproject.org/; see also Supplementary Table 7). Source data are provided with this paper.

## Code availability
All code used for data processing and analysis in this study is publicly available on GitHub (https://github.com/jknightlab/MTOR-Genetics-Project).

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

## Acknowledgements

This project was supported by the Medical Research Council (MR/V002503/1) [J.C.K.], a Wellcome Trust Investigator Award [204969/Z/16/Z] [J.C.K.], the Chinese Academy of Medical Sciences (CAMS) Innovation Fund for Medical Science (CIFMS), China (grant number: 2024-I2M-2-001-1) [P.Z., J.C.K.], Wellcome Trust Grants [090532/Z/09/Z and 203141/Z/16/Z to core facilities Centre for Human Genetics, and the NIHR Oxford Biomedical Research Centre. Computation used the Oxford Biomedical Research Computing (BMRC) facility, a joint development between the Wellcome Centre for Human Genetics and the Big Data Institute, supported by Health Data Research UK and the NIHR Oxford Biomedical Research Centre. The Wellcome Sanger Institute is funded by the Wellcome Trust [220540/Z/20/A]. C.O.N. is supported by a Wellcome Trust Doctorate Award (228321/Z/23/Z). A.J.M. received support from the Academy of Medical Sciences Starter Grant (SGL024∖1096). A.J.M., J.O.J.D., and J.C.K. received support from the National Institute for Health Research (NIHR) Oxford Biomedical Research Centre (BRC). For the purpose of Open Access, the authors have applied a CC BY public copyright licence to any Author Accepted Manuscript arising from this submission.

## Author contributions

P.Z. and J.C.K. conceived the study. P.M., A.J., and I.D. curated the clinical datasets for the GAinS and SI cohorts. K.L.B. and E.E.D. prepared the genotype and gene expression data for GAinS. C.O.N., A.Y.C., and A.J.M. curated the genotype and clinical data for the GenoSept and UKB cohorts. C.W., Q.H., and G.S. coordinated patient recruitment, sample collection, and lab support. A.A. prepared the RNA-seq and ATAC-seq samples from T cells. P.Z., E.P., and J.C.K. designed the co-culture experiments. P.Z., J.C.K., and J.O.J.D. designed the CBE base-editing strategy. B.O. and R.D. cloned the CBE plasmids and synthesised the mRNAs. K.C.G. and A.C.B. provided the software tools and resources. P.Z. performed the data analysis and modelling. P.Z. and J.C.K. interpreted the results and drafted the manuscript with input from all authors.

## Competing interests

J.C.K. reports a grant to his institution from the Danaher Beacon Programme for work on RNA biomarker point-of-care test development in sepsis for endotype assignment, which includes support for K.C.-G. and J.C.K. All remaining authors declare that they have no competing interests.
