## [Transparent Peer Review file · Nature Communications]

Context-specific regulatory genetic variation in MTOR dampens neutrophil-T cell crosstalk in pneumonia-associated sepsis

Corresponding Author: Dr Ping Zhang

Version 0:

Reviewer comments:

Reviewer #1

(Remarks to the Author)

The study by Zhang et al. employed a robust genetic analysis to elucidate the association between the MTOR eQTL variant rs4845987 and both immunological and clinically relevant parameters. In sepsis, the minor G-allele of rs4845987 was associated with an increased expression of MTOR, particularly in the SRS1 endotype, and interacted with neutrophil-to-lymphocyte ratio (NLR). The minor G-allele was further associated with reduced mortality in patients with sepsis due to community-acquired pneumonia (CAP) and with decreased susceptibility to type 2 diabetes (T2D). Mechanistically, the Authors revealed a dichotomous role of mTOR signaling in T lymphocytes and neutrophils. In particular, mTOR modulated the T cell-dependent induction of sepsis-associated neutrophil phenotypes, as indicated by the regulation of CD64, CD123, and PD-L1 expression. Through integrative analysis of epigenomic datasets combined with CRISPR/dCas9-mediated editing in T cells, the Authors showed direct evidence that the eQTL variant rs4845987 maps to an enhancer critically involved in the epigenetic regulation of MTOR.

Overall, the study provides valuable insights into the contribution of the MTOR eQTL to sepsis immunopathogenesis and prognosis, and sheds light on the epigenetic regulation of the locus, as well as the interplay between neutrophils and T cells. However, some conclusions are insufficiently supported or appear contradictory, warranting further demonstration and explanation by the Authors. In addition, methodological concerns require clarification.

1) Cell-type deconvolution highlighted the key role of neutrophils, including the IL-1R2 immature subset, within the SRS1 endotype. Among the inferred immune populations, non-classical monocytes emerged as relevant (Fig. 1c), and a significant effect of the minor-G allele was observed in monocytes (Fig. 1f). Dysregulated myelopoiesis and induction of IL-1R2 immune-suppressive monocytes with prognostic significance have been reported in sepsis across multiple independent studies. By focusing solely on neutrophils, the study overlooks the relevance of MTOR eQTLs on monocyte frequency and function. This limitation should be addressed and/or subjected to a more critical discussion in the manuscript.

2) In support of the study, the Authors highlighted the increased susceptibility to pneumonitis associated with mTOR inhibitor treatment in cancer patients (Albiges et al., *Ann Oncol*, 2012). In addition, in patients with T2D, who are susceptible to sepsis, the Authors emphasized that sodium-glucose cotransporter-2 inhibitors suppress mTOR activity and reduce pneumonia risk (Henney et al., *Thorax*, 2024; Fukushima et al., *Frontiers in Pharmacology*, 2021). The dichotomous role of mTOR, along with the contrasting effects of its pharmacological inhibition (i.e., increased pneumonia risk in cancer patients versus decreased susceptibility in individuals with T2D) may pose significant interpretative challenges for readers. Therefore, the Authors are encouraged to directly investigate, where possible, the influence of the rs4845987 allele on pneumonia risk in T2D and cancer patients.

3) Neutrophil activation was assessed by evaluating the surface expression of CD64, CD123 and PD-L1, which are upregulated in sepsis as shown by the Authors in Fig. S4b. However, in Fig. 3b, the thresholds used to define CD64+, CD123+ and PD-L1+ neutrophils (dotted lines) appeared arbitrary and were not established using unbiased internal controls, such as Fluorescence Minus One (FMO) or isotype staining. Notably, using the threshold established in Figure 3b, the Authors did not observe upregulation of CD64, CD123, and PD-L1 on neutrophils from either the sepsis patient or the healthy donor depicted in the histograms. It is therefore recommended that the Authors also report the mean or median fluorescence intensity (MFI) values, which would provide a more accurate and quantitative representation of protein levels in this experimental setting. In the bar plot of Fig. 4a, it would be beneficial to represent the relative percentage of CD69+PD-1+ lymphocytes within the "activated T cells", thereby underscoring the suppressive effects of neutrophils in co-culture.

Likewise, the relative percentage of CD123+ CD64+ neutrophils in the absence of T cells should be included in the bar plot of Fig. 4b.

4) The statement that CD64, CD123 and PD-L1 were upregulated at mRNA level in SRS1 compared to non-SRS1 endotype (Fig. S4b) should be substantiated by quantitative analysis including aggregated data from all patients. Similarly, the distinctive effects of CD4+ and CD8+ T cells on neutrophils in Fig. S4c should be better formalized displaying CD64+ and CD123+ cells in all patients analyzed.

5) The authors demonstrated the critical role of activated T cells in the rapamycin-mediated regulation of neutrophil phenotype (Fig. S4). It would be valuable to further interpret this evidence with an emphasis on its translational implications.

6) Concerns arise regarding the gating strategy presented in Fig. S6b. The criteria employed by the Authors to distinguish CD66b⁺ from CD66b⁻ cells should be explained, as this marker exhibits low expression levels comparable to those observed in CD3+ lymphocytes. Clarification is needed on how potential contaminants (i.e., monocytes, B cells) were excluded from the whole blood analysis. The absence of labels on both the x- and y-axes compromises the interpretability of the figure.

7) The minor G-allele was associated with reduced 28-day mortality in CAP sepsis patients and, based on this observation, the Author interpreted it as consistent with the role of mTOR in hypoxia and oxidative stress. In addition, the epigenetic regulation of MTOR prompted the manipulation of hydroxymethylation using vitamin C (VC), with reported increases in the MTOR expression. To strengthen the manuscript, these findings could be more coherently harmonized, especially considering the potential detrimental effects of VC therapy in sepsis, as hypothesized. Providing direct evidence that VC modulates hydroxymethylation at the MTOR locus would be particularly valuable. Furthermore, the Authors should substantiate the impact of VC on T cell-dependent activation of sepsis neutrophils (i.e., expression of surface markers, NETosis), under at least part of the relevant experimental conditions (i.e., normoxia versus hypoxia, following treatment with rapamycin).

(Remarks on code availability)

Reviewer #2

(Remarks to the Author)

This is a fascinating manuscript by Zhang et al, connecting NLR to sepsis outcomes via the mTOR pathway. Specifically, the authors demonstrate that the G-allele of rs4845987 interacts with the sepsis marker NLR and exhibits activity specific to the sepsis endotype. In addition, they also show that the G-allele rs4845987 is associated with decreased risk of T2D, associated with decreased mTOR signaling in T cells, and improved sepsis survival in pneumonia patients. However, there appears to be a disconnect between the studies on T2D and those on sepsis. The question is, can they demonstrate differences in sepsis patients with T2D compared to those without, based on the G-allele and/or NLR?

(Remarks on code availability)

Reviewer #3

(Remarks to the Author)

The manuscript titled "Context-specific regulatory genetic variation in MTOR dampens neutrophil-T cell crosstalk in sepsis, modulating disease" presents a compelling investigation into the intersection of genetic variation, immune regulation, and clinical outcomes in sepsis. The study is innovative in its integration of genomic, epigenetic, immunologic, and clinical datasets to elucidate how the rs4845987 variant modulates MTOR expression and influences disease progression and survival in sepsis. Notably, the identification of a cell-type-specific eQTL with opposing regulatory effects in T cells and neutrophils is a particularly striking finding that offers promising avenues for precision medicine approaches in the management of sepsis. That said, while the overall scientific contribution is substantial, the manuscript needs critical clarifications and improvements in several areas, particularly regarding mechanistic causality and the design and interpretation of the ex vivo neutrophil co-culture experiments. Below are detailed comments and suggestions to strengthen the manuscript:

Major comments:

1) Although the authors demonstrate a statistically significant association between the rs4845987 G-allele and reduced MTOR expression in activated T cells (Figure 1f), the conclusion that this variant functionally reduces mTOR signaling remains largely inferential. Could the authors provide direct functional evidence that the G-allele causally leads to reduced mTOR signaling, beyond eQTL or correlative data? The authors could consider: a) performing allele-specific CRISPR editing in T cells to introduce the G or C allele at the endogenous locus and measure effects on MTOR expression and downstream signaling (e.g., p-S6, p-4EBP1); or b) using enhancer reporter assays with the rs4845987-containing sequence (G vs. C allele) to directly test transcriptional activity in activated T cells.

2) In the ex vivo co-culture experiments (Methods; Figure S5C), neutrophils from septic patients were incubated with T cells in the presence of rapamycin, and readout was evaluated at 2 or 4 days. Given that neutrophils have a minimal lifespan ex vivo, it is unclear how the authors accounted for neutrophil viability at the 4-day time point. Furthermore, septic neutrophils are known to be pre-activated and functionally dysregulated, making them more prone to spontaneous apoptosis or NETosis even in short-term culture. The readout observed at later time points may be influenced by neutrophil death processes, particularly NETosis, rather than representing true immunomodulatory effects from T cells or rapamycin. This is especially relevant at 4 days, when neutrophils are unlikely to remain viable, and NETs or cell debris may confound interpretation. Please provide data on neutrophil viability and NETosis (e.g., Annexin V/PI, SYTOX Green, or MPO-DNA ELISA). Given

their pre-activated state, these cells may have a shorter functional lifespan *ex vivo* compared to healthy controls. If viability is compromised, results should be interpreted with caution or omitted. Alternatively, consider repeating key assays with shorter incubation periods or incorporating time-course analyses that correlate phenotypic changes with cell survival status.

3) In Figure S4A, the gating strategy used to distinguish neutrophils and T cells in co-culture experiments is presented. However, the plots do not indicate whether gating on live cells was performed prior to downstream analysis (e.g., surface marker expression such as CD64, CD123, PD-L1, or CD69). Without explicit gating on viable cells, the analysis may include apoptotic or dead cells, which could artificially influence marker expression and skew interpretation. This is particularly relevant in the context of: Neutrophils, which rapidly undergo apoptosis or NETosis *ex vivo*. Co-cultures performed over extended durations (e.g., 48–96 hours). Please clarify whether a live/dead cell discriminator (e.g., Zombie Aqua, PI, 7-AAD) was included in the flow cytometry panel. If not, we recommend incorporating viability gating in all future analyses and explicitly showing this step in updated gating plots. If viability gating was performed but not shown, please update Figure S4A to reflect the full gating strategy, including the initial exclusion of dead cells.

4) In Figure S4C, the gating strategy used to identify CD64+ and CD123+ neutrophils is not clearly defined. The contour plots appear to show a broad smear or gradient of signal, without clear demarcation of a distinct positive population. This makes it difficult to assess whether the reported shifts in marker expression represent true induction in a defined neutrophil subset, or simply a diffuse increase in background staining or autofluorescence. Without a well-defined positive gate, comparisons across conditions may be unreliable. Please provide a more rigorous gating strategy, potentially incorporating additional neutrophil markers (e.g., CD15, CD16, CD66b, or SSC properties) to delineate the population of interest clearly. Show Fluorescence Minus One (FMO) controls or clearly defined thresholds to distinguish positive vs. negative cells. Consider adding quantitative histograms or overlay plots to illustrate shifts in marker expression more effectively, rather than relying solely on 2D plots with diffuse populations.

5) While the authors report a statistically significant association between the rs4845987 G-allele and improved survival in patients with sepsis due to community-acquired pneumonia (CAP) across multiple cohorts, this observation remains correlational. Could the authors provide additional evidence supporting a causal role of this variant in modulating disease outcome? For example, is there functional validation, such as gene editing or allele-specific manipulation, that demonstrates a direct contribution of rs4845987 to immune regulation or survival pathways? Furthermore, since the protective effect is restricted to CAP-associated sepsis and not observed in non-CAP patients, the title and abstract should be revised to specify the pulmonary origin of sepsis. For example: pneumonia-associated sepsis. This change would prevent overgeneralization.

6) The study relies heavily on association-based data (eQTL mapping, survival correlations, colocalization) without fully demonstrating causality. Although the CRISPRa experiments and vitamin C assays are valuable, they do not directly confirm whether the observed survival effects are mediated through mTOR transcriptional modulation *in vivo*. Consider including *in vivo* models (e.g., humanized mice) to strengthen causal inference.

7) The involvement of epigenetic mechanisms, specifically hydroxymethylation and the role of TET enzymes, is introduced relatively late in the manuscript, which limits its integration into the broader mechanistic narrative. The authors propose that vitamin C (VC) increases mTOR expression in activated T cells in an rs4845987 genotype-dependent manner, mediated by enhancer activity and TET enzyme activation. However, this conclusion remains primarily associative and lacks direct experimental evidence linking VC treatment to allele-specific epigenetic modulation at the mTOR locus. To strengthen this claim, the manuscript would benefit from clearer functional or temporal linkage between enhancer accessibility, 5hmC deposition, and transcriptional activation of mTOR. Specifically, the authors could demonstrate a) allele-specific enrichment of 5hmC following VC exposure; b) differential recruitment of TET enzymes at the enhancer site; and c) discussion of the potential clinical relevance of VC administration in sepsis patients, particularly in C/C carriers, who may experience deleterious mTOR overactivation.

8) The concept of stratifying sepsis patients based on endotype (SRS1 vs. non-SRS1) and NLR levels is powerful, but its clinical implementation remains underexplored. Expand discussion on how rs4845987 genotyping or mTOR-related biomarkers could be used in the clinic. Is there potential for patient selection in trials of rapamycin, TET inhibitors, or vitamin C?

Minor comments:

9) The schematic presented in Figure 5i appears to be duplicated in the graphical abstract, conveying essentially the same conclusion and visual summary. To avoid redundancy, it is recommended to retain only one of these images. The Graphical Abstract is likely more suitable for summarizing the findings at a glance. Therefore, please consider removing Figure 5i, or alternatively, replacing it with a complementary schematic that adds new mechanistic insight.

10) Throughout the manuscript, there is an inconsistency in the terminology used to describe T cell activation states: Some figures use the terms “Rest” vs. “Activated”, which clearly indicates resting versus stimulated T cells. Other figures use “Total” vs. “Activated”, which creates confusion. It is unclear whether “Total” refers to resting cells, a mixture of activation states, or another specific subset. Please standardize the terminology across the manuscript and figures. If “Total” is equivalent to “Resting,” using only “Resting” vs. “Activated” would enhance clarity. If “Total” represents a distinct population, it should be clearly defined in both the figure legends and the main text.

11) Consider including a supplemental table mapping all significant eQTLs that interact with NLR or SRS1 status for transparency.

(Remarks on code availability)

Reviewer #4

(Remarks to the Author)

The objective of the studies conducted by Dr. Zhang and colleagues was to characterize genetic variation that modulates MTOR in the setting of sepsis. The manuscript first identifies and characterizes an MTOR eQTL that impacts expression differently in activated T-cells and neutrophils, shows activity that is specific to sepsis endotypes, and impacts sepsis prognosis marker neutrophil to lymphocyte ratio. Next, the manuscript reports the findings in ex vivo models of the interaction of T-cells and neutrophils from septic patients and healthy controls, including experiments in the presence of rapamycin and hypoxia. The findings have implications for future sepsis therapeutics targeting MTOR pathways with a precision medicine approach. The studies are detailed, well thought through, and well executed. The manuscript is overall well written, although can be confusing to follow and the relevance of every experiment is not immediately apparent. Overall, I find the studies reported to be significant and well conducted. I have a few concerns listed below.

Major Comments:

1. The timing of blood collection in the human cohorts for gene expression and neutrophil to lymphocyte ratio relative to sepsis/critical illness onset is likely important as these outcomes are likely highly dynamic early in sepsis. Do the authors have information regarding the relative timing across the different cohorts?
2. Additionally, neutrophil to lymphocyte ratio is a relatively crude measure of the interaction of neutrophils and lymphocytes. Other crude measures such as absolute neutrophil and lymphocyte count may be important as well as the relative concentrations of circulating inflammatory mediators/cytokines. This limitation should be acknowledged.
3. The manuscript gets confusing with regards to the direction of effects. The GG allele results in increased MTOR expression in whole blood which results in improved survival. But later experiments report increased T-cell expression of MTOR is associated with poorer outcomes. It seems from the data that this is due to differential effects between neutrophil MTOR expression and T-cell MTOR expression. If this is the case, can the authors make this clearer in the manuscript, possibly with a figure, and provide an explanation as to why this may be the case.
4. In section 2 of the results, why do the authors believe that the non-SRS1 patients rather than the SRS1 patients have improved survival if they possess the G-allele when the G-allele is a stronger eQTL in the SRS1 group. Wouldn't one hypothesize that the effect should be stronger in the SRS1 group?
5. The comparison of the associations of genotype with mortality stratified by pneumonia vs non-pneumonia are interesting but hypothesis generating as stratified analyses suggest this to be true but statistical interaction testing is not provided.
6. How was hospital acquired pneumonia handled in the stratified analyses? Were they included with community acquired pneumonia or non-pneumonia?
7. It is unclear the relevance of the shared MTOR genetic variant with sepsis and T2D. This section is underdeveloped and may not have relevance to the mechanisms proposed in the rest of the experiments.
8. The discussion section should include a discussion of the limitations of the studies as they do not prove, only suggest, mechanism.

(Remarks on code availability)

Reviewer #5

(Remarks to the Author)

This manuscript investigates genetic variation influencing MTOR, a key regulator of metabolism and immune responses in sepsis. The authors propose that the effects of this variant are highly context-specific, mediated by a regulatory element that modulates MTOR expression in activated T cells and neutrophils with opposing effects. The study is supported by both in vivo and in vitro models. It addresses an important question regarding the genetic determinants of sepsis, particularly in SRS1, a subset of patients with more severe disease, and has significant implications for precision-targeted therapies. The experimental design is generally rigorous, and the study is well-conceived. However, it has several major weaknesses that limit the strength of its conclusions, along with a few minor issues that could be resolved through clarification and revision.

Major points:

1. The manuscript lacks a description of key population characteristics, including sex, age, race/ethnicity, treatment variables (e.g., steroid or immunomodulatory therapy, timing of antibiotic administration), and time-related variables (e.g., time from symptom onset to sample collection). In addition, among the 638 sepsis patients, it is unclear how many were classified as SRS1 and how many as non-SRS1. The criteria used to define SRS1 versus non-SRS1 should be explicitly stated. Including a schematic or workflow figure may also help clarify the study design and analysis pipeline.
2. When the authors investigate the interaction effects of SRS endotypes on the MTOR eQTL association in whole blood (Fig. 1b), it is important to clarify whether this represents an SRS-specific (response) eQTL observed during sepsis, or a constitutive eQTL that is also present in whole blood of healthy populations. The current analysis only shows that MTOR eQTLs are absent in naïve CD4+ and CD8+ T cells, but it remains unclear whether such eQTLs are present in whole blood or other immune cell types, such as B cells, monocytes, and others, in both septic and healthy conditions. Additional clarification or data on MTOR eQTL status across these contexts would strengthen the interpretation.
3. The manuscript does not specify how many cell types were included in the deconvolution analysis. Additionally, it is

unclear how the authors handled nested cell populations, for example, both CD8+ T cells (parent) and naïve CD8+ T cells (offspring) present in the deconvoluted cell types. This is important to clarify, as absolute proportions derived from statistical deconvolution typically sum to one, and including both parent and subset populations may lead to overlapping or ambiguous estimates. It would be helpful to include supplementary boxplots comparing the subject level proportions of each deconvoluted cell type between the SRS1 and non-SRS1 groups, to enhance interpretation of cell-type-specific differences across endotypes.

4. When using PEER factor analysis, it is unclear whether the deconvoluted cell type proportions were included as known covariates or whether blood composition was treated as a hidden variable. Additionally, the manuscript does not clarify whether all covariates, including PEER factors, were regressed out to generate residuals for downstream analyses, or whether they were incorporated directly as covariates in regression models. Clarifying the modeling strategy is important for interpreting how potential confounding by cell type composition and technical variation was addressed.

5. The sample size ($n = 823$) exceeds the number of subjects ($n = 638$), suggesting that some individuals contributed more than one sample. This should be explicitly stated and clarified in the Methods. If repeated measures are present, the use of a linear mixed-effects model is appropriate to account for within-subject correlation. However, the manuscript should clearly describe the study design (e.g., longitudinal sampling or multiple conditions per subject) and justify the choice of statistical model accordingly.

Minor points:

6. What is the minor allele frequency of rs4845987?

7. Fig. 1c. Non-classical monocytes and S100A8_9_hi neutrophils which has higher Beta coefficient? It would be helpful to indicate this more clearly in the figure.

8. Table S2 requires clearer annotation. For example, it is unclear whether “Blood” refers to whole blood or PBMCs. Additionally, abbreviations such as “HLC” and others should be defined either in the table legend or as footnotes to improve interpretability.

9. Line 241 missed a period after the word crucial.

(Remarks on code availability)

Version 1:

Reviewer comments:

Reviewer #1

(Remarks to the Author)

The Authors have extensively revised the manuscript, demonstrating scientific rigor and attention to each point raised. The methodology is convincing, and the point-to-point responses are comprehensive and well justified. Both the strengths and the corresponding limitations of the manuscript are clearly articulated. The revised version further enhances the quality and general impact of a very intriguing study.

(Remarks on code availability)

Reviewer #2

(Remarks to the Author)

This is a resubmission of a very interesting manuscript, which is highly clinically significant to the field. The work supports the conclusions, and adequate evidence is provided. The methodology is sound. The authors have answered all the queries and changed the manuscript accordingly.

(Remarks on code availability)

Reviewer #3

(Remarks to the Author)

Follow-up comment (on Major Comment)

I appreciate the authors' additional data on neutrophil viability and their clarification regarding the co-culture experiments. However, I remain concerned about the experimental design and the biological relevance of the time points used.

First, the inclusion of day 4 neutrophil data is not appropriate, as neutrophils are extremely short-lived cells that rarely maintain functionality beyond 24 hours ex vivo. The authors themselves report only ~25% viability at day 4, indicating that the vast majority of cells are dead or undergoing apoptosis/NETosis. Data derived from such cultures are unlikely to represent true neutrophil-T cell interactions and should therefore be removed entirely from the manuscript and supplementary materials.

Second, even at day 2, viability is only ~70–75%, which remains relatively low for reliable interpretation of immune crosstalk. At this stage, dying cells and NETotic debris can substantially affect cytokine levels, activation markers, and readouts

attributed to “functional” interactions.

To strengthen the physiological relevance of these assays, I strongly recommend that the authors consider performing shorter co-cultures (e.g., 12–18 hours), which would better reflect viable neutrophil–T cell communication. Such experiments would help confirm whether the observed effects truly arise from active crosstalk rather than secondary consequences of neutrophil death.

If shorter co-cultures cannot be performed at this stage, the authors should explicitly acknowledge this limitation in the Discussion and state clearly that results obtained at or beyond 2 days must be interpreted with caution due to progressive neutrophil apoptosis and loss of function.

(Remarks on code availability)

no

Reviewer #4

(Remarks to the Author)

I have read the revised manuscript and believe the authors have done an excellent job addressing the reviewers concerns. I have no further comments.

(Remarks on code availability)

Reviewer #5

(Remarks to the Author)

The authors have addressed my previous questions, and I have no additional comments.

(Remarks on code availability)

Version 2:

Reviewer comments:

Reviewer #3

(Remarks to the Author)

My previous questions have been addressed, and I have no further comments.

(Remarks on code availability)

yes, Remarks on code availability.

REVIEWER COMMENTS

Reviewer #1 (Remarks to the Author):

The study by Zhang et al. employed a robust genetic analysis to elucidate the association between the MTOR eQTL variant rs4845987 and both immunological and clinically relevant parameters. In sepsis, the minor G-allele of rs4845987 was associated with an increased expression of MTOR, particularly in the SRS1 endotype, and interacted with neutrophil-to-lymphocyte ratio (NLR). The minor G-allele was further associated with reduced mortality in patients with sepsis due to community-acquired pneumonia (CAP) and with decreased susceptibility to type 2 diabetes (T2D). Mechanistically, the Authors revealed a dichotomous role of mTOR signaling in T lymphocytes and neutrophils. In particular, mTOR modulated the T cell-dependent induction of sepsis-associated neutrophil phenotypes, as indicated by the regulation of CD64, CD123, and PD-L1 expression. Through integrative analysis of epigenomic datasets combined with CRISPR/dCas9-mediated editing in T cells, the Authors showed direct evidence that the eQTL variant rs4845987 maps to an enhancer critically involved in the epigenetic regulation of MTOR.

Overall, the study provides valuable insights into the contribution of the MTOR eQTL to sepsis immunopathogenesis and prognosis, and sheds light on the epigenetic regulation of the locus, as well as the interplay between neutrophils and T cells. However, some conclusions are insufficiently supported or appear contradictory, warranting further demonstration and explanation by the Authors. In addition, methodological concerns require clarification.

We thank the reviewer for this valuable feedback and recognition of our work. We have addressed the concerns regarding the conclusions and methodology in detail below.

1) Cell-type deconvolution highlighted the key role of neutrophils, including the IL-1R2 immature subset, within the SRS1 endotype. Among the inferred immune populations, non-classical monocytes emerged as relevant (Fig. 1c), and a significant effect of the minor-G allele was observed in monocytes (Fig. 1f). Dysregulated myelopoiesis and induction of IL-1R2 immune-suppressive monocytes with prognostic significance have been reported in sepsis across multiple independent studies. By focusing solely on neutrophils, the study overlooks the relevance of MTOR eQTLs on monocyte frequency and function. This limitation should be addressed and/or subjected to a more critical discussion in the manuscript.

We thank the reviewer for this helpful suggestion. We agree that dysregulated myelopoiesis is critical in the progression of sepsis, and indeed, similar to NLR, the monocyte-to-lymphocyte ratio (MLR) has been reported as a useful marker to identify high-risk septic patients. We therefore tested the interaction between MLR and the *MTOR* eQTL in the GAinS cohort. In contrast to NLR, we did not observe a significant interaction of MLR with the eQTL association (revised **Supplementary Fig. 1i**), nor did we detect a significant independent detrimental effect of MLR on 28-day sepsis mortality (revised **Supplementary Fig. 2b**), suggesting that bulk monocyte-T cell crosstalk may be dispensable for the *MTOR*-dependent effect during sepsis. However, we cannot rule out the contribution of other cell subsets, such as non-classical monocytes (as highlighted by the reviewer) and/or monocytes under endotoxin tolerance. Indeed, our previous studies demonstrate that prolonged LPS stimulation (24-hour) induces tolerance in monocytes, where the eQTL association for the *MTOR* SNP was also observed. We have therefore added the following to the revised Discussion section: *“Previous work indicates that immune suppressive monocytes show prognostic significance in sepsis^{1,2} and that prolonged endotoxin exposure induces tolerance in monocytes^{3,4}, a setting in which the MTOR eQTL effect was also observed. This further underscores the importance of considering cell-type-specific and subset-specific regulatory mechanisms. We demonstrated allele-specific epigenomic regulation in T cells,*

but the molecular mechanisms underlying the inverted effect in neutrophils and monocytes remain unclear. Future studies incorporating single-cell epigenomics, cytokine profiling and humanised mouse models will be critical to dissect the functional heterogeneity of immune cell populations and to define the cellular and molecular mechanisms underlying MTOR-dependent genetic regulation and complex immune interactions.” on page 14 (lines 345–354) of the revised manuscript.

2) In support of the study, the Authors highlighted the increased susceptibility to pneumonitis associated with mTOR inhibitor treatment in cancer patients (Albiges et al., Ann Oncol, 2012). In addition, in patients with T2D, who are susceptible to sepsis, the Authors emphasized that sodium-glucose cotransporter-2 inhibitors suppress mTOR activity and reduce pneumonia risk (Henney et al., Thorax, 2024; Fukushima et al., Frontiers in Pharmacology, 2021). The dichotomous role of mTOR, along with the contrasting effects of its pharmacological inhibition (i.e., increased pneumonia risk in cancer patients versus decreased susceptibility in individuals with T2D) may pose significant interpretative challenges for readers. Therefore, the Authors are encouraged to directly investigate, where possible, the influence of the rs4845987 allele on pneumonia risk in T2D and cancer patients.

We thank the reviewer for this insightful comment. We previously highlighted the increased susceptibility to pneumonitis associated with mTOR inhibitor treatment in cancer patients (Albiges et al., Ann Oncol, 2012). However, our new data show that neutrophil markers responsive to T cell activation were upregulated in sepsis, but not in a sterile inflammation control group (patients undergoing cardiac surgery) (**Supplementary Fig. 4h**). This indicates that the MTOR genetic association, and the resulting neutrophil–T cell interaction observed in our study, is more likely to reflect pathogen-driven immune responses rather than sterile inflammation. We have therefore removed the text “*which is consistent with the role of mTOR in hypoxia/oxidative stress and evidence linking mTOR inhibitors to pneumonitis*” in the revised version of the manuscript.

Using UK Biobank data, we stratified pneumonia patients according to T2D status, baseline HbA1c levels (a biomarker of T2D), and all-cause cancer registry-derived cases prior to sepsis onset. It is known that many cancers create an immunosuppressive environment characterised by exhausted T cells. Consistent with stratification by immunosuppressed status (IS), we observed a significant protective effect of the G allele in non-cancer patients (revised **Supplementary Fig. 2c-d**). We further observed significant allelic effects in patients without T2D or cancers in both the UK Biobank and GAinS sepsis cohorts (revised **Supplementary Fig. 2c-d**), suggesting that T cell dysfunction in T2D may mask the eQTL effect and neutrophil–T cell crosstalk. Supporting this, a recent study demonstrated that sustained hyperglycemia, as seen in T2D, can impair T cell signaling and metabolism ⁵.

To further explore this and minimise the possibility that the observed effect was due to sample size or power limitations, we stratified UK Biobank patients by HbA1c levels (not available in GAinS). As expected, HbA1c was higher in T2D patients (revised **Supplementary Fig. 2e**). Consistently, the protective effect of the G allele on survival was observed among patients with lower HbA1c (revised **Supplementary Fig. 2d** in purple).

We have included these above results as a new supplementary **figure S2c-e** and detailed the analysis in the Methods Section. We have also added the following text on page 5 (lines 150-157) of the revised manuscript: “*Consistent with the immune paresis nature of the SRS1 endotype, the protective effect of the G-allele in UK Biobank patients (Fig. 2c) was observed only in those without an immunosuppressed status (IS) and without cancers (Fig. 2e, Supplementary Fig. 2c-d; see Methods). Stratification by T2D and HbA1c (a T2D marker; Supplementary Fig. 2e) further showed that this protective effect was lost in T2D patients and in those with higher HbA1c levels (Supplementary Fig. 2c-d). In line with this, a recent*

study demonstrated that sustained hyperglycemia, as seen in T2D, can impair T cell signalling and metabolism⁵.

3) Neutrophil activation was assessed by evaluating the surface expression of CD64, CD123 and PD-L1, which are upregulated in sepsis as shown by the Authors in Fig. S4b. However, in Fig. 3b, the thresholds used to define CD64+, CD123+ and PD-L1+ neutrophils (dotted lines) appeared arbitrary and were not established using unbiased internal controls, such as Fluorescence Minus One (FMO) or isotype staining. Notably, using the threshold established in Figure 3b, the Authors did not observe upregulation of CD64, CD123, and PD-L1 on neutrophils from either the sepsis patient or the healthy donor depicted in the histograms. It is therefore recommended that the Authors also report the mean or median fluorescence intensity (MFI) values, which would provide a more accurate and quantitative representation of protein levels in this experimental setting. In the bar plot of Fig. 4a, it would be beneficial to represent the relative percentage of CD69+PD-1+ lymphocytes within the “activated T cells”, thereby underscoring the suppressive effects of neutrophils in co-culture. Likewise, the relative percentage of CD123+ CD64+ neutrophils in the absence of T cells should be included in the bar plot of Fig. 4b.

We thank Reviewer for these helpful suggestions. In the original Fig. 3b, we applied the threshold for neutrophils relative to the culture without T cells as shown in Fig. 2b for quantification. We have now exported the median fluorescence intensity (MFI) values using flowJo, and compared the differences of these markers between sepsis and healthy. We observed significant upregulation of the proteins on sepsis neutrophils relative to HVs at both base level and upon co-culture with T cells. We have added this revised **Fig.3c** in the new version of our manuscript.

We also added a flow cytometry histogram showing the relative expression of these surface markers on neutrophils relative to the corresponding isotope controls in **Supplementary Fig. 4b** in our revised manuscript. We have changed the Y axis labels for **Fig.3d-e** as ‘Neutrophils relative to culture without T cells’ to clarify the control that we used for the comparisons.

We have now plotted the relative percentage of CD69+PD-1+ lymphocytes within the “activated T cells” in the revised **bar plot of Fig.4a**, and included the relative percentage of CD123+ CD64+ neutrophils in the absence of T cells in the revised **bar plot of Fig.4b**.

4) The statement that CD64, CD123 and PD-L1 were upregulated at mRNA level in SRS1 compared to non-SRS1 endotype (Fig. S4b) should be substantiated by quantitative analysis including aggregated data from all patients. Similarly, the distinctive effects of CD4+ and CD8+ T cells on neutrophils in Fig. S4c should be better formalized displaying CD64+ and CD123+ cells in all patients analyzed.

We thank reviewer for this helpful suggestion. After quantifying the transcription of these markers on neutrophils using aggregated scRNA-seq data from all patients, we observed significant upregulation of all markers in sepsis patients compared to healthy donors and more profound differences with severe SRS1 patients (see revised **Supplementary Fig. 4g**). This upregulation is specific to sepsis, as no significant difference was observed between healthy donors and patients after cardiac surgery (a sterile inflammation control group) (revised **Supplementary Fig. 4h**).

We also observed an elevated expression of FCGR1A (encoding CD64) in SRS1 versus non-SRS1 patients, and a similar but not statistically significant trend for the other two genes likely due to limited statistical power. We have added these new figure panels as **Supplementary Fig. 4g-h** in the revised version of our manuscript and removed the

following conclusion in the main text '*The expression levels were further elevated in SRS1 compared to non-SRS1 patients*'.

For the original Supplementary Fig. 4c, we have added the flow cytometry histogram showing the expression of markers in primary CD4⁺ and CD8⁺ T cell along with immobilised Jurkat cells (revised **Supplementary Fig. 4c**). We have also quantified the MFI values of these markers on neutrophils derived from different patients (see revised **Supplementary Fig. 4d**; n=5 for CD4⁺ co-culture; n=4 for CD8⁺ co-culture; n=6 for Jurkat). We see significant upregulation of neutrophil marker expression co-cultured with activated cells relative to resting cells for both CD4⁺ and CD8⁺ T cells, but no effect was observed from Jurkat. We have added these new supplementary figure panels **Supplementary Fig. 4c-d** in our revised manuscript.

5) The authors demonstrated the critical role of activated T cells in the rapamycin-mediated regulation of neutrophil phenotype (Fig. S4). It would be valuable to further interpret this evidence with an emphasis on its translational implications.

We thank the reviewer for this valuable suggestion. We have added the following sentence to the first paragraph of the Discussion section (page 14; line 336-339) of our revised manuscript: "*Our data demonstrate selective modulation of excessive mTOR signaling in T cells by rapamycin alleviates neutrophil hyperactivation, potentially restoring immune balance without broad immunosuppression and highlighting T cell-neutrophil crosstalk as a therapeutic target in sepsis due to pneumonia.*"

6) Concerns arise regarding the gating strategy presented in Fig. S6b. The criteria employed by the Authors to distinguish CD66b⁺ from CD66b⁻ cells should be explained, as this marker exhibits low expression levels comparable to those observed in CD3⁺ lymphocytes. Clarification is needed on how potential contaminants (i.e., monocytes, B cells) were excluded from the whole blood analysis. The absence of labels on both the x- and y-axes compromises the interpretability of the figure.

We thank reviewer for this insightful comment. To address these concerns, we first performed an additional analysis using whole blood from 3 sepsis patients incubated with conditioned media from 5 healthy donors. A new box plot (revised **Supplementary Fig. 6d**) has been included to quantify expression across replicates. We have also revised **Supplementary Fig. 6b** showing the labels on both the x- and y-axis. We have added these figure panels as **Supplementary Fig. 6b-d** in our revised manuscript and updated the legend and main text.

To exclude the potential contaminants of monocytes and B cells, we used antibodies against CD14-AF700 (Biolegend, #367114) and CD19-APC (Biolegend, #302212) to gate the monocytes and B cells respectively. In contrast to CD66b⁺ neutrophils (**Fig.R2a**), we did not observe upregulation of CD64 and PD-L1 expression on these cells derived from the same sepsis patient in the presence of conditioned media from activated T cells (**Fig.R2b-d**). We saw subtle upregulation of CD123 in monocytes but not in B cells (**Fig.R2c**). We agree that the interaction between monocytes and T cells is critical in many infections. Accordingly, we have added a discussion on this point on page 14 (lines 341-354). The added discussion highlights that prolonged endotoxin exposure induces tolerance in monocytes, where the *MTOR* eQTL effect is also observed, emphasising cell type- and subset-specific regulatory mechanisms.

Fig. R2. (a) Flow cytometry histograms showing upregulation of CD64, CD123 and PD-L1 on CD66b⁺ sepsis neutrophils cultured with conditioned media from activated T cells of 4 healthy donors. T cells were stimulated with anti-CD3/CD28 beads for 2 or 3 days. (b) Gating strategy for CD14⁺ monocytes and CD19⁺ B cells. (c-d) Flow cytometry histograms showing upregulation of CD64, CD123, and PD-L1 on CD14⁺ monocytes (c) and CD19⁺ B cells (d) from the same sepsis patient shown in (a).

7) The minor G-allele was associated with reduced 28-day mortality in CAP sepsis patients and, based on this observation, the Author interpreted it as consistent with the role of mTOR in hypoxia and oxidative stress. In addition, the epigenetic regulation of MTOR prompted the manipulation of hydroxymethylation using vitamin C (VC), with reported increases in the MTOR expression. To strengthen the manuscript, these findings could be more coherently harmonized, especially considering the potential detrimental effects of VC therapy in sepsis, as hypothesized. Providing direct evidence that VC modulates hydroxymethylation at the MTOR locus would be particularly valuable. Furthermore, the Authors should substantiate the impact of VC on T cell-dependent activation of sepsis neutrophils (i.e., expression of surface markers, NETosis), under at least part of the relevant experimental conditions (i.e., normoxia versus hypoxia, following treatment with rapamycin).

We thank the reviewer for these helpful suggestions. To investigate the impact of VC on T cell-dependent activation, we performed MTOR knockdown using siRNAs. We achieved ~40% reduction in MTOR transcription using three independent siRNAs in activated T cells derived from healthy donors (revised **Supplementary Fig. 9a**). This reduction resulted in a significant decrease in IFN- γ release both at the basal level and following VC treatment (revised **Fig.5g**), consistent with the RNA-seq analysis showing rapamycin-dependent regulation of IFN- γ transcription as shown in Fig.4g. Moreover, we also observed a significant reduction in MTOR transcription by mutating the C allele (associated with increased MTOR expression and higher sepsis mortality) at its endogenous locus using base editing. For this, we utilised a Cas9 mutant (SpG) with an expanded PAM recognition site (NGN) fused to a high-efficiency cytidine deaminase, enabling precise editing of the MTOR SNP in primary T cells (revised **Fig.5j-l**).

We have included these new data in Figure panels **Supplementary Fig. 9a-b and Fig.5g, j, k and l**, and a new Results paragraph (page 11-12) and a detailed Methods section describing the construct design, cell delivery and functional assays (page 20) in our revised manuscript. In addition, we have added a new paragraph to discuss the effect of VC and outlining future directions, including the potential importance of the identified genetic variant as a biomarker for disease severity, and how hydroxymethylation dynamics and TET inhibition could represent promising therapeutic targets for sepsis due to pneumonia on page 15 of the revised manuscript.

Reviewer #2 (Remarks to the Author):

This is a fascinating manuscript by Zhang et al, connecting NLR to sepsis outcomes via the mTOR pathway. Specifically, the authors demonstrate that the G-allele of rs4845987 interacts with the sepsis marker NLR and exhibits activity specific to the sepsis endotype. In addition, they also show that the G-allele rs4845987 is associated with decreased risk of T2D, associated with decreased mTOR signaling in T cells, and improved sepsis survival in pneumonia patients.

However, there appears to be a disconnect between the studies on T2D and those on sepsis. The question is, can they demonstrate differences in sepsis patients with T2D compared to those without, based on the G-allele and/or NLR?

We thank the reviewer for recognizing the significance of our study and for raising the valuable question regarding stratification by T2D status in sepsis patients.

We stratified pneumonia patients according to T2D status, using baseline HbA1c levels (a biomarker of T2D) prior to sepsis onset using the UK Biobank data. We observed significant allelic effects in patients without T2D in both the UK Biobank and GAInS sepsis cohorts (revised **Supplementary Fig. 2c-d**), suggesting that T cell dysfunction in T2D may mask the eQTL effect and neutrophil-T cell crosstalk. Supporting this, a recent study demonstrated that sustained hyperglycemia, as seen in T2D, can impair T cell signaling and metabolism (Gray V et al., Cell Metab. 2024; PMID:39488214). To further explore this and minimise the possibility that the observed effect was due to sample size or power limitations, we stratified UK Biobank patients by HbA1c levels (not available in GAInS). As expected, HbA1c was higher in T2D patients (revised **Supplementary Fig. 2e**). Consistently, the protective effect of the G allele on survival was observed among patients with lower HbA1c (revised **Supplementary Fig. 2d** with purple dots).

We have included these above results in our revised manuscript as follows: “*Consistent with the immune paresis nature of the SRS1 endotype, the protective effect of the G-allele in UK Biobank patients (Fig. 2c) was observed only in those without an immunosuppressed status (IS) and without cancers (Fig. 2e, Supplementary Fig. 2c-d; see Methods). Stratification by T2D and HbA1c (a T2D marker; Supplementary Fig. 2e) further showed that this protective effect was lost in T2D patients and in those with higher HbA1c levels (Supplementary Fig. 2c-d). In line with this, a recent study demonstrated that sustained hyperglycemia, as seen in T2D, can impair T cell signalling and metabolism.*⁵” on page 5, and detailed the method for UKB data stratification on page 17.

Reviewer #3 (Remarks to the Author):

The manuscript titled "Context-specific regulatory genetic variation in MTOR dampens

neutrophil-T cell crosstalk in sepsis, modulating disease" presents a compelling investigation into the intersection of genetic variation, immune regulation, and clinical outcomes in sepsis. The study is innovative in its integration of genomic, epigenetic, immunologic, and clinical datasets to elucidate how the rs4845987 variant modulates MTOR expression and influences disease progression and survival in sepsis. Notably, the identification of a cell-type-specific eQTL with opposing regulatory effects in T cells and neutrophils is a particularly striking finding that offers promising avenues for precision medicine approaches in the management of sepsis. That said, while the overall scientific contribution is substantial, the manuscript needs critical clarifications and improvements in several areas, particularly regarding mechanistic causality and the design and interpretation of the ex vivo neutrophil co-culture experiments. Below are detailed comments and suggestions to strengthen the manuscript:

We are grateful to the reviewer for their recognition of the study's significance and for providing helpful feedback to improve the manuscript. We have revised the manuscript and supporting information to address the comments.

Major comments:

1) Although the authors demonstrate a statistically significant association between the rs4845987 G-allele and reduced MTOR expression in activated T cells (Figure 1f), the conclusion that this variant functionally reduces mTOR signaling remains largely inferential. Could the authors provide direct functional evidence that the G-allele causally leads to reduced mTOR signaling, beyond eQTL or correlative data? The authors could consider: a) performing allele-specific CRISPR editing in T cells to introduce the G or C allele at the endogenous locus and measure effects on MTOR expression and downstream signaling (e.g., p-S6, p-4EBP1); or b) using enhancer reporter assays with the rs4845987-containing sequence (G vs. C allele) to directly test transcriptional activity in activated T cells.

We thank the reviewer for these helpful suggestions. To directly assess the allelic effect of the prioritised eQTL SNP rs4845987 on *MTOR* expression, we utilised a base-editing approach to introduce a C-to-T substitution at the endogenous locus⁶. We co-delivered SpG-TadCBE6b mRNA, a Cas9 mutant with an expanded PAM recognition site (NGN) fused to a high-efficiency cytidine deaminase, together with sgRNAs precisely targeting this variant in activated T cells with the C/C genotype (revised **Fig. 5j**; see Methods). We observed efficient on-target C-to-T conversion at both the SNP site (revised **Fig. 5k**) and a control site located 100bp upstream (revised **Supplementary Fig. 9b**). Editing at the SNP site resulted in an ~20% reduction in MTOR expression, compared with editing of the upstream control region or non-edited cells. A minor bystander C-to-T conversion was detected 5 bp downstream of the SNP. However, this cytosine lacks any known regulatory or functional relevance and is therefore unlikely to contribute to the observed effect (revised **Supplementary Fig. 9c**), as discussed on page 14 (line 356-364) of our revised manuscript.

Additionally, to further enhance MTOR transcription downregulation and determine its effect on downstream signalling, we knocked down MTOR transcription (~40% reduction) using independent siRNAs (revised **Supplementary Fig. 9a**). This reduction resulted in a significant decrease in IFN- γ release, both at baseline and following VC treatment (revised **Fig. 5g**), consistent with the RNA-seq analysis showing rapamycin-dependent regulation of IFN- γ transcription as shown in Fig. 4g. These results align with the pro-inflammatory role of IFN- γ -secreting T cells^{7,8}, and with recent randomised clinical trial data suggesting potential harmful effects of IFN- γ therapy in hospital-acquired pneumonia cases⁹.

We have included these new results in Figure panels **Supplementary Fig. 9a-b** and **Fig. 6g, j-l**, and added a Methods section describing the construct design, cell delivery, and functional assays on page 20 of the revised manuscript.

2) In the ex vivo co-culture experiments (Methods; Figure S5C), neutrophils from septic patients were incubated with T cells in the presence of rapamycin, and readout was evaluated at 2 or 4 days. Given that neutrophils have a minimal lifespan ex vivo, it is unclear how the authors accounted for neutrophil viability at the 4-day time point. Furthermore, septic neutrophils are known to be pre-activated and functionally dysregulated, making them more prone to spontaneous apoptosis or NETosis even in short-term culture. The readout observed at later time points may be influenced by neutrophil death processes, particularly NETosis, rather than representing true immunomodulatory effects from T cells or rapamycin. This is especially relevant at 4 days, when neutrophils are unlikely to remain viable, and NETs or cell debris may confound interpretation. Please provide data on neutrophil viability and NETosis (e.g., Annexin V/PI, SYTOX Green, or MPO-DNA ELISA). Given their pre-activated state, these cells may have a shorter functional lifespan ex vivo compared to healthy controls. If viability is compromised, results should be interpreted with caution or omitted. Alternatively, consider repeating key assays with shorter incubation periods or incorporating time-course analyses that correlate phenotypic changes with cell survival status.

We thank reviewer for these comments. The 4-day treatment shown in **Supplementary Fig. 5c (right panel)** was designed to assess the dose/time effects and potential toxicity of rapamycin. Based on this assessment, we therefore selected the 2 days co-culture with 100nM rapamycin for the results presented in **Supplementary Fig. 5b and Fig.3d**. We have now empathised this in the corresponding legend, and the Methods section to include: "Co-culture was performed for 2 days unless otherwise indicated."

In 2 days, we observed on average ~70% viable neutrophils across different patient samples (revised **Fig.5f** red bars). Neutrophil viability was decreased to 25% in 4 days (revised **Fig.5f** cyan bars), however increasing concentrations of rapamycin had no observable effect on neutrophils viability in both 2 days and 4 days (revised **Fig.5f**), suggesting this dose-dependent effect of rapamycin did not affect neutrophil viability. We have added these new panels (**Supplementary Fig. 5e-f**) showing the gating and percentage of live/dead neutrophils across treatments at both day 2 and day 4 in our revised manuscript. NETosis was quantified after 4~6 hours of incubation with Cytotox Green, following the manufacturer's protocol (Sartorius). Objects <300 μm^2 were excluded, as recommended to distinguish apoptotic cells from NETosis. We have also included these details in the corresponding figure legend of our revised manuscript.

3) In Figure S4A, the gating strategy used to distinguish neutrophils and T cells in co-culture experiments is presented. However, the plots do not indicate whether gating on live cells was performed prior to downstream analysis (e.g., surface marker expression such as CD64, CD123, PD-L1, or CD69). Without explicit gating on viable cells, the analysis may include apoptotic or dead cells, which could artificially influence marker expression and skew interpretation. This is particularly relevant in the context of: Neutrophils, which rapidly undergo apoptosis or NETosis ex vivo. Co-cultures performed over extended durations (e.g., 48–96 hours). Please clarify whether a live/dead cell discriminator (e.g., Zombie Aqua, PI, 7-AAD) was included in the flow cytometry panel. If not, we recommend incorporating viability gating in all future analyses and explicitly showing this step in updated gating plots. If viability gating was performed but not shown, please update Figure S4A to reflect the full gating strategy, including the initial exclusion of dead cells.

We thank the reviewer for this comment. We used an amine-reactive dye for live cell staining and have now highlighted this in the gating strategy of **Supplementary Fig. 4a**. We have

updated the corresponding figure legend to include the following: “Viable neutrophils were gated using LIVE/DEAD™ Fixable Green Dead Cell Stain Kit (ThermoFisher)”.

4) In Figure S4C, the gating strategy used to identify CD64+ and CD123+ neutrophils is not clearly defined. The contour plots appear to show a broad smear or gradient of signal, without clear demarcation of a distinct positive population. This makes it difficult to assess whether the reported shifts in marker expression represent true induction in a defined neutrophil subset, or simply a diffuse increase in background staining or autofluorescence. Without a well-defined positive gate, comparisons across conditions may be unreliable. Please provide a more rigorous gating strategy, potentially incorporating additional neutrophil markers (e.g., CD15, CD16, CD66b, or SSC properties) to delineate the population of interest clearly. Show Fluorescence Minus One (FMO) controls or clearly defined thresholds to distinguish positive vs. negative cells. Consider adding quantitative histograms or overlay plots to illustrate shifts in marker expression more effectively, rather than relying solely on 2D plots with diffuse populations.

We thank the reviewer for this comment. We have highlighted the gating strategy in **Supplementary Fig. 4a**, including SSC properties, an amine-reactive LIVE/DEAD dye and neutrophil marker CD66b+ to subset neutrophils from T cells for quantifications. We have used matched isotope controls to define the positive signals of neutrophils surface markers, which has been included as **Supplementary Fig. 4b** in the revised version of the manuscript. We have added the histograms, and reported the median fluorescence intensity (MFI) values derived from different individuals, to provide a more accurate and quantitative marker expression (revised **Supplementary Fig. 4c-d**).

5) While the authors report a statistically significant association between the rs4845987 G-allele and improved survival in patients with sepsis due to community-acquired pneumonia (CAP) across multiple cohorts, this observation remains correlational. Could the authors provide additional evidence supporting a causal role of this variant in modulating disease outcome? For example, is there functional validation, such as gene editing or allele-specific manipulation, that demonstrates a direct contribution of rs4845987 to immune regulation or survival pathways? Furthermore, since the protective effect is restricted to CAP-associated sepsis and not observed in non-CAP patients, the title and abstract should be revised to specify the pulmonary origin of sepsis. For example: pneumonia-associated sepsis. This change would prevent overgeneralization.

We thank the reviewer for these helpful suggestions. To investigate the impact of gene editing or allele-specific manipulation of *MTOR* expression on immune regulation, we first performed gene knockdown using siRNAs. We achieved ~40% reduction in *MTOR* transcription using three independent siRNAs in activated T cells derived from healthy donors (revised **Supplementary Fig. 9a**). This reduction resulted in a significant decrease in IFN- γ release both at the basal level and following VC treatment (revised **Fig. 5g**), consistent with the RNA-seq analysis showing rapamycin-dependent regulation of IFN- γ transcription as shown in Fig. 4g. Next, to directly determine the allelic effect of the prioritised eQTL SNP rs4845987 on *MTOR* expression, as described above for addressing comment 1, we utilised a base-editing approach to introduce a C-to-T substitution at the endogenous locus⁶. We cloned and co-delivered SpG-TadCBE6b mRNA (a Cas9 mutant with an expanded PAM recognition site (NGN) fused to a high-efficiency cytidine deaminase) and sgRNAs precisely targeting the variant in activated T cells with C/C genotype (revised **Fig. 5j**; see **Methods**). Efficient C-to-T conversions were observed at both the SNP site (revised **Fig. 5k**) and a control site located ~100 bp upstream (revised **Supplementary Fig. 9b**). We found that editing at the SNP resulted in a ~20% reduction in *MTOR* expression compared with editing of the upstream control region or non-edited cells (revised **Fig. 5l**). A minor bystander C-to-T

conversion was detected 5 bp downstream of the SNP (revised **Fig. 5k**), but this cytosine lacks any known regulatory or functional association (revised **Supplementary Fig. 9c**).

We have included above new results in Figure panels **Supplementary Fig. 9a-b and Fig. 5g, j, k and l**, a Results paragraph and a Methods section describing the construct design, cell delivery and functional assays on page 11-12 and 20 of the revised manuscript, respectively.

We have clarified in the title, abstract and throughout the main text that the protective effect of this genetic variant on sepsis survival is specific to pneumonia. In the revised Discussion section (page 14; lines 341-345), we have discussed on the potential molecular mechanisms underlying this association.

6) The study relies heavily on association-based data (eQTL mapping, survival correlations, colocalization) without fully demonstrating causality. Although the CRISPRa experiments and vitamin C assays are valuable, they do not directly confirm whether the observed survival effects are mediated through MTOR transcriptional modulation in vivo. Consider including in vivo models (e.g., humanized mice) to strengthen causal inference.

We thank the reviewer for this helpful comment. We have utilised both gene-editing and base-editing approaches to address this, as detailed in our responses to Comments 1 and 5. We agree that a humanised mouse model would be an excellent approach to investigate the complex immune interactions, and we have included this point in our revised Discussion section.

7) The involvement of epigenetic mechanisms, specifically hydroxymethylation and the role of TET enzymes, is introduced relatively late in the manuscript, which limits its integration into the broader mechanistic narrative. The authors propose that vitamin C (VC) increases MTOR expression in activated T cells in an rs4845987 genotype-dependent manner, mediated by enhancer activity and TET enzyme activation. However, this conclusion remains primarily associative and lacks direct experimental evidence linking VC treatment to allele-specific epigenetic modulation at the MTOR locus. To strengthen this claim, the manuscript would benefit from clearer functional or temporal linkage between enhancer accessibility, 5hmC deposition, and transcriptional activation of MTOR. Specifically, the authors could demonstrate a) allele-specific enrichment of 5hmC following VC exposure; b) differential recruitment of TET enzymes at the enhancer site; and c) discussion of the potential clinical relevance of VC administration in sepsis patients, particularly in C/C carriers, who may experience deleterious mTOR overactivation.

We thank the reviewer for the insightful suggestions. To functionally link MTOR regulation with allele-specific and epigenetic mechanisms, we first performed siRNA-mediated knockdown in activated T cells from healthy donors, achieving ~40% reduction in MTOR transcripts and a corresponding decrease in IFN- γ release both basally and after VC treatment (revised **Supplementary Fig. 9a, Fig. 5g**). This supports our RNA-seq finding of rapamycin-dependent IFN- γ regulation (Fig. 4g). To directly assess the allelic effect of rs4845987, we applied base editing to introduce a C-to-T substitution at the endogenous locus using SpG-TadCBE6b and variant-specific sgRNAs (revised **Fig. 5j**). Efficient C-to-T conversion was achieved at the SNP and a nearby control site (~100 bp upstream; **Fig. 5k, S9b**). Editing the SNP reduced MTOR expression by ~20% compared to control or unedited cells (Fig. 5l). These results, now included in revised **Figs. S9a-b and 5g, j-l**, with corresponding Results and Methods updates, provide direct evidence for allele-specific modulation of MTOR expression, the reduction of which shows VC-dependent key cytokine release, supporting potential VC- and TET-dependent epigenetic regulation which we have now discussed in details in the last paragraph on page 15. Specifically, we have added a new paragraph discussing the effects of vitamin C and outlining future directions, highlighting

the potential importance of the identified genetic variant as a biomarker for disease severity and emphasizing that further investigation of 5hmC dynamics and TET inhibition may reveal promising therapeutic targets for pneumonia-associated sepsis.

8) The concept of stratifying sepsis patients based on endotype (SRS1 vs. non-SRS1) and NLR levels is powerful, but its clinical implementation remains underexplored. Expand discussion on how rs4845987 genotyping or mTOR-related biomarkers could be used in the clinic. Is there potential for patient selection in trials of rapamycin, TET inhibitors, or vitamin C?

We thank the Reviewer for this helpful suggestion. In response, we have added a new paragraph to the Discussion section (page 15) of our revised manuscript, expanding on how rs4845987 genotyping and mTOR-related biomarkers could inform clinical implementation. Specifically, we now discuss the potential mechanisms underlying the harmful effects of VC treatment and the implications for patient stratification in future trials of rapamycin and TET inhibitors.

Minor comments:

9) The schematic presented in Figure 5i appears to be duplicated in the graphical abstract, conveying essentially the same conclusion and visual summary. To avoid redundancy, it is recommended to retain only one of these images. The Graphical Abstract is likely more suitable for summarizing the findings at a glance. Therefore, please consider removing Figure 5i, or alternatively, replacing it with a complementary schematic that adds new mechanistic insight.

Thank you. We have removed Figure 5i.

10) Throughout the manuscript, there is an inconsistency in the terminology used to describe T cell activation states: Some figures use the terms “Rest” vs. “Activated”, which clearly indicates resting versus stimulated T cells. Other figures use “Total” vs. “Activated”, which creates confusion. It is unclear whether “Total” refers to resting cells, a mixture of activation states, or another specific subset. Please standardize the terminology across the manuscript and figures. If “Total” is equivalent to “Resting,” using only “Resting” vs. “Activated” would enhance clarity. If “Total” represents a distinct population, it should be clearly defined in both the figure legends and the main text.

Thank you. We have corrected this in the revised manuscript and ensured consistency in using the terms “resting” and “activated.”

11) Consider including a supplemental table mapping all significant eQTLs that interact with NLR or SRS1 status for transparency.

Thank you. We have included the genome-wide lead eQTLs and their interactions with NLR and SRS status in Supplementary Table S1 of the revised manuscript.

Reviewer #4 (Remarks to the Author):

The objective of the studies conducted by Dr. Zhang and colleagues was to characterize genetic variation that modulates MTOR in the setting of sepsis. The manuscript first identifies and characterizes an MTOR eQTL that impacts expression differently in activated T-cells and neutrophils, shows activity that is specific to sepsis endotypes, and impacts sepsis prognosis marker neutrophil to lymphocyte ratio. Next, the manuscript reports the

findings in ex vivo models of the interaction of T-cells and neutrophils from septic patients and healthy controls, including experiments in the presence of rapamycin and hypoxia. The findings have implications for future sepsis therapeutics targeting MTOR pathways with a precision medicine approach. The studies are detailed, well thought through, and well executed. The manuscript is overall well written, although can be confusing to follow and the relevance of every experiment is not immediately apparent. Overall, I find the studies reported to be significant and well conducted. I have a few concerns listed below.

We thank the reviewer for these constructive feedback and recognition of the significance of our study. We have revised the manuscript and supporting information to address the comments.

Major Comments:

1. The timing of blood collection in the human cohorts for gene expression and neutrophil to lymphocyte ratio relative to sepsis/critical illness onset is likely important as these outcomes are likely highly dynamic early in sepsis. Do the authors have information regarding the relative timing across the different cohorts?

We agree with the reviewer that gene expression and NLR are highly dynamic during acute infection and play an important role in determining clinical outcomes. In the Genomic Advances in Sepsis (GAInS) and Sepsis Immunomics (SI) studies, blood samples were collected serially after hospital admission for gene expression profiling (where possible on days 1, 3 and 5 after admission). However, gene expression and NLR data were not available for patients from the UK Biobank (UKB) and GenOSept cohorts.

For the association test between the SNP and 28-day mortality, we used two complementary approaches to assign gene expression profiles to Sepsis Response Signature (SRS) status: 1. SRS-latest (assigned based on samples from the latest available time point for each patient); 2. SRS1-ever (assigned if any sample from a patient was classified as SRS1 at any time point). Importantly, the association results as shown Fig. 2d remained robust across both assignment methods. To maximise sample size and proximity to the clinical endpoint, we used the latest available NLR measurements for analysis.

2. Additionally, neutrophil to lymphocyte ratio is a relatively crude measure of the interaction of neutrophils and lymphocytes. Other crude measures such as absolute neutrophil and lymphocyte count may be important as well as the relative concentrations of circulating inflammatory mediators/cytokines. This limitation should be acknowledged.

We thank the reviewer for this helpful comment. We agree that the NLR is a relatively crude measure of immune cell interactions and may not fully reflect their dynamic interplay. We have now acknowledged this limitation in the Discussion section (page 14-15) and noted that future studies incorporating single-cell epigenomic and cytokine profiling will be critical to dissect the functional heterogeneity of immune cell populations and to define the cellular and molecular mechanisms underlying MTOR-dependent genetic regulation and complex immune interactions.

3. The manuscript gets confusing with regards to the direction of effects. The GG allele results in increased MTOR expression in whole blood which results in improved survival. But later experiments report increased T-cell expression of MTOR is associated with poorer outcomes. It seems from the data that this is due to differential effects between neutrophil MTOR expression and T-cell MTOR expression. If this is the case, can the authors make this clearer in the manuscript, possibly with a figure, and provide an explanation as to why this may be the case.

We thank the reviewer for this helpful comment. We agree that the direction of the MTOR effect may appear confusing at first. Indeed, we observed an inverse effect of the G allele on

MTOR expression between neutrophils and activated T cells: the G allele is associated with increased MTOR expression in neutrophils, but reduced MTOR expression in activated T cells carrying the same allele. Our survival analyses across multiple cohorts consistently demonstrated a protective effect of the G allele on sepsis survival, among patients with non-SRS endotypes, without immunosuppression, and without pre-existing immunological conditions such as type 2 diabetes or cancer, both of which are known to impair T-cell function. Together with our functional experiments, these findings support a model in which excessive T-cell activation and cytokine release during sepsis drive hyperactivation of neutrophils, disrupting immune homeostasis and leading to poor clinical outcomes. We have clarified this throughout the manuscript and included a schematic model (Fig. 6) illustrating the stimulation-specific effects of MTOR and its effect on immune regulation and survival.

4. In section 2 of the results, why do the authors believe that the non-SRS1 patients rather than the SRS1 patients have improved survival if they possess the G-allele when the G-allele is a stronger eQTL in the SRS1 group. Wouldn't one hypothesize that the effect should be stronger in the SRS1 group?

We thank the reviewer for this helpful comment. We agree that based on the stronger eQTL effect of the G allele in whole blood from SRS1 patients, we might initially expect a greater survival benefit in this group. However, the SRS1 subgroup is characterised by a higher abundance of neutrophils, where the G allele increases MTOR expression, but this eQTL effect is significantly reduced in non-SRS1 patients and in those with a lower neutrophil-to-lymphocyte ratio (NLR) as shown in Fig. 1b and 1d, suggesting an important interplay between neutrophils and T cells in modulating the overall effect of the variant. Consistent with this hypothesis, we observed an inverse effect of the G allele on MTOR expression in activated T cells, indicating that the G allele may help balance MTOR signalling during sepsis. When this balance is disrupted in patients with immune suppression and impaired T-cell activity, the protective effect of the G allele is lost. This hypothesis is consistent with our survival analyses using multiple cohorts and stratified analysis, as described above and clarified in the revised manuscript.

5. The comparison of the associations of genotype with mortality stratified by pneumonia vs non-pneumonia are interesting but hypothesis generating as stratified analyses suggest this to be true but statistical interaction testing is not provided.

We thank the reviewer for this helpful comment and agree that this pneumonia-specific genetic association is an interesting and hypothesis-generating observation. Our stratified analyses suggest that the effect may be specific to sepsis due to pneumonia. However, we currently lack direct molecular evidence to explain this phenotype. We did not observe a significant interaction when fitting a Cox proportional hazards model including a genotype x diagnosis term adjusted for age and sex. This is likely due to limited statistical power in the non-CAP group, which had substantially fewer events (deaths) (10.7%; 41 out of 384), compared with the CAP group (18.2%; 134 of 737).

We have therefore acknowledged this limitation and proposed the pneumonia-specific association as a hypothesis for future investigation in the Discussion section (Page 14-15) of the revised manuscript. In addition, we now discuss the possibility that tissue-resident immune interactions in the lung may be central to mediating this organ-specific genetic effect. Dysregulated neutrophil activation and crosstalk with adaptive immune cells could amplify local inflammation and influence the impact of MTOR-dependent pathways on sepsis outcomes in pneumonic infection.

6. How was hospital acquired pneumonia handled in the stratified analyses? Were they included with community acquired pneumonia or non-pneumonia?

Thank you. Patients with hospital acquired pneumonia were not included in the stratified analyses as community-acquired pneumonia (CAP) cases. In the GAInS cohort, only patients with CAP or faecal peritonitis (non-CAP) were recruited. We have now highlighted the CAP definition in the Methods section as follows: “Community-acquired pneumonia (CAP) was diagnosed as a febrile illness with cough, sputum production, breathlessness, leukocytosis and radiological evidence of pneumonia, acquired in the community or within 2 days of hospital admission.”

7. It is unclear the relevance of the shared MTOR genetic variant with sepsis and T2D. This section is underdeveloped and may not have relevance to the mechanisms proposed in the rest of the experiments.

Thank you for this helpful comment. As addressed in our responses to Reviewers 1 and 3, to test the potential relevance of the shared MTOR genetic variant with sepsis and T2D, we stratified pneumonia patients according to T2D status, baseline HbA1c levels (a biomarker of T2D) prior to sepsis onset using UK Biobank data. We further observed significant allelic effects in patients without T2D or cancers in both the UK Biobank and GAInS sepsis cohorts (revised **Supplementary Fig. 2c-d**), suggesting that T cell dysfunction in T2D may mask the eQTL effect and neutrophil-T cell crosstalk. Supporting this, a recent study demonstrated that sustained hyperglycemia, as seen in T2D, can impair T cell signalling and metabolism ⁵.

To further explore this and minimise the possibility that the observed effect was due to sample size or power limitations, we stratified UK Biobank patients by HbA1c levels (not available in GAInS). As expected, HbA1c was higher in T2D patients (revised **Supplementary Fig. 2e**). Consistently, the protective effect of the G allele on survival was observed among patients with lower HbA1c (revised **Supplementary Fig. 2d** in purple). We have included these new results in our revised manuscript and discussed the sepsis-T2D interaction in Discussion.

8. The discussion section should include a discussion of the limitations of the studies as they do not prove, only suggest, mechanism.

In the revised Discussion section, we now acknowledge the limitations of the study: (1) Base editing using TadCBEs may introduce bystander edits outside the canonical 4-8 nt window, representing a technical limitation that could be minimised through the use of narrower editors (e.g., TadCBE_d) or PAM-relaxed Cas9 variants (e.g., SpRY). (2) Our analyses were performed at the bulk level, which may obscure cell type-specific heterogeneity that could be resolved through future single-cell approaches. (3) Future studies integrating single-cell epigenomic and cytokine profiling, together with humanized mouse models, will be critical to dissect the functional heterogeneity of immune cell populations and to define the cellular and molecular mechanisms underlying MTOR-dependent genetic regulation, as well as genotype-dependent interactions between 5hmC, TET activity, and immune cell subsets.

Reviewer #5 (Remarks to the Author):

This manuscript investigates genetic variation influencing MTOR, a key regulator of metabolism and immune responses in sepsis. The authors propose that the effects of this variant are highly context-specific, mediated by a regulatory element that modulates MTOR expression in activated T cells and neutrophils with opposing effects. The study is supported by both in vivo and in vitro models. It addresses an important question regarding the genetic determinants of sepsis, particularly in SRS1, a subset of patients with more severe disease, and has significant implications for precision-targeted therapies. The experimental design is generally rigorous, and the study is well-conceived. However, it has several major

weaknesses that limit the strength of its conclusions, along with a few minor issues that could be resolved through clarification and revision.

We are grateful to the reviewer for their thoughtful and positive assessment. We have revised the manuscript and supporting information accordingly to address all comments and suggestions.

Major points:

1. The manuscript lacks a description of key population characteristics, including sex, age, race/ethnicity, treatment variables (e.g., steroid or immunomodulatory therapy, timing of antibiotic administration), and time-related variables (e.g., time from symptom onset to sample collection). In addition, among the 638 sepsis patients, it is unclear how many were classified as SRS1 and how many as non-SRS1. The criteria used to define SRS1 versus non-SRS1 should be explicitly stated. Including a schematic or workflow figure may also help clarify the study design and analysis pipeline.

We thank the reviewer for these valuable suggestions. We have added demographic and sample collection information in the Methods sections "Survival analysis" and "UK Biobank (UKB) data curation and analysis", and have cited the relevant publications describing the detailed cohort characteristics. Among the 638 sepsis patients, 36.7% (234 out of 638) and 44.4% (283 out of 638) were classified as SRS1 using the SRS-latest and SRS1-ever assignments, respectively. Specifically: SRS-latest was assigned based on samples from the latest available time point for each patient; SRS1-ever was assigned if any sample from a patient was classified as SRS1 at any time point. We have added the analysis pipeline on GitHub, and also clarified the criteria used to define SRS1 on page 16 (Line 412-415) of our revised manuscript as follows: "SRS and SRSq scores were estimated using a 7-gene signature implemented through Sepstratifier, as previously described. Among the 638 GAInS patients, 36.7% and 44.4% were classified as SRS1 using the SRS-latest and SRS1-ever assignments respectively."

2. When the authors investigate the interaction effects of SRS endotypes on the MTOR eQTL association in whole blood (Fig. 1b), it is important to clarify whether this represents an SRS-specific (response) eQTL observed during sepsis, or a constitutive eQTL that is also present in whole blood of healthy populations. The current analysis only shows that MTOR eQTLs are absent in naïve CD4+ and CD8+ T cells, but it remains unclear whether such eQTLs are present in whole blood or other immune cell types, such as B cells, monocytes, and others, in both septic and healthy conditions. Additional clarification or data on MTOR eQTL status across these contexts would strengthen the interpretation.

We thank the reviewer for this helpful comment. To clarify, our analysis focused on eQTLs with strong colocalization signals (posterior probability, $PP4 > 0.95$) across multiple datasets. The MTOR eQTL signal showed high colocalisation ($PP4 > 0.95$) with MTOR expression in activated T cells, neutrophils, and other cell types, including tolerised monocytes and adipose tissue, as shown in Fig.1f and Table S2. In addition, this eQTL colocalised with whole blood data from GTEx, Lepik et al., 2017 and TwinsUK, with $PP4$ values of 0.94, 0.73, and 0.56 respectively (Table S2), suggesting that this is a constitutive eQTL detectable in healthy conditions but functionally modulated under sepsis. We did not observe significant eQTL association in B cells from healthy ($PP4 < 0.2$ across three datasets; Table S2), and no B-cell data during sepsis were available for further validation. We have clarified these points in the Discussion section (page 14, lines 341-354), where we now describe the context-dependent modulation of the MTOR eQTL across immune cell types and its potential activation-specific behaviour during sepsis.

3. The manuscript does not specify how many cell types were included in the deconvolution analysis. Additionally, it is unclear how the authors handled nested cell populations, for

example, both CD8+ T cells (parent) and naïve CD8+ T cells (offspring) present in the deconvoluted cell types. This is important to clarify, as absolute proportions derived from statistical deconvolution typically sum to one, and including both parent and subset populations may lead to overlapping or ambiguous estimates. It would be helpful to include supplementary boxplots comparing the subject level proportions of each deconvoluted cell type between the SRS1 and non-SRS1 groups, to enhance interpretation of cell-type-specific differences across endotypes.

Thank you for this helpful comment. The cell annotations in the original the sepsis whole blood scRNA-seq dataset¹⁰ are not nested but represent distinct clusters at the same annotation level. For deconvolution, we included all cell types except apoptosing cells (n=23) as described in this dataset. Regarding the annotated cell population, “CD8+ T cells” and “naïve CD8+ T cells” have distinct expression profiles (see below **Fig.R3a-b**), and the “CD8+ T cells” show higher expression of key activation markers GZMB and IFNG, and lower expression of naïve markers CCR7 (see below **Fig.R3c-d**). To clarify this distinction, we therefore renamed the cluster as “CD8+ T cells (activated)” for visualisation purposes in the figures of our revised manuscript. We have added supplementary boxplots (revised **Supplementary Fig. 10**) comparing the sample-level absolute scores of each deconvoluted cell type between SRS1 and non-SRS1 groups, and revised the corresponding Methods on page 16 as follows: “*CIBERSORTx absolute scores for each cell type in bulk RNA-seq samples were estimated using the bulk count matrix and a signature matrix of 23 distinct cell subsets excluding apoptosing cells derived from sepsis scRNA-seq data, with S-mode batch correction and 100 permutations. Sample-level absolute scores across subsets were compared between SRS1 and non-SRS1 groups using a linear mixed model accounting for donor variability (Supplementary Fig. 10).*”

Fig. R3. (a-c) UMAP projection of the Kwok et al. whole blood sepsis scRNA-seq dataset¹⁰ coloured for CD8+ T cells (a-b), and for the expression of activation markers GZMB and IFNG, and the naïve cell marker CCR7 (c). (d) Boxplots showing the transcript levels of GZMB, IFNG and CCR7 in this dataset. Normalised expression values were averaged across cells per sample (sample-level aggregates). Group comparisons were tested using two-tailed t-tests. ***P<0.001.

4. When using PEER factor analysis, it is unclear whether the deconvoluted cell type proportions were included as known covariates or whether blood composition was treated as a hidden variable. Additionally, the manuscript does not clarify whether all covariates,

including PEER factors, were regressed out to generate residuals for downstream analyses, or whether they were incorporated directly as covariates in regression models. Clarifying the modeling strategy is important for interpreting how potential confounding by cell type composition and technical variation was addressed.

We thank the reviewer for highlighting the need for greater clarity regarding our data processing steps. 1. eQTL mapping and interaction analysis: We used gene expression data corrected for population structure and PEER factors to maximise power for eQTL association testing, as described in the Methods (page 15-16; line 405-415). 2. Cell-type deconvolution: We applied DESeq2 normalisation to the raw bulk RNA-seq count matrix, without regressing out blood-composition covariates, in order to preserve the native cell-composition signal within our whole blood samples for accurate fraction estimation using CIBERSORTx and a signature matrix derived from sepsis whole blood single-cell RNAseq (page 16; line 422-430). 3. Survival and regression analyses: All models explicitly accounted for key clinical and technical covariates, including age, sex, and genetic structure PCs as detailed in the corresponding figure legends.

5. The sample size ($n = 823$) exceeds the number of subjects ($n = 638$), suggesting that some individuals contributed more than one sample. This should be explicitly stated and clarified in the Methods. If repeated measures are present, the use of a linear mixed-effects model is appropriate to account for within-subject correlation. However, the manuscript should clearly describe the study design (e.g., longitudinal sampling or multiple conditions per subject) and justify the choice of statistical model accordingly.

Thank you. We have clarified in the Methods section that some individuals contributed multiple longitudinal samples, resulting in a total of 823 samples from 638 patients. To account for inter-individual variability, we used a linear mixed-effects model, specifying donor ID as a random effect (page 16; line 411-415).

Minor points:

6. What is the minor allele frequency of rs4845987?

We thank the reviewer for this comment. The minor G-allele frequency of rs4845987 is 0.39 in global populations, and 0.91, 0.31, 0.21, 0.30 and 0.38 in Africans, Americans, East Asians, Europeans and South Asians, respectively. This information has been included in the revised version of our manuscript.

7. Fig. 1c. Non-classical monocytes and S100A8_9_hi neutrophils which has higher Beta coefficient? It would be helpful to indicate this more clearly in the figure.

Thank you. The S100A8_9_hi neutrophil subset has the higher Beta coefficient, and this is now indicated more clearly in Fig. 1c with arrows.

8. Table S2 requires clearer annotation. For example, it is unclear whether “Blood” refers to whole blood or PBMCs. Additionally, abbreviations such as “HLC” and others should be defined either in the table legend or as footnotes to improve interpretability.

Thank you. We have changed “blood” to “whole blood” and added the abbreviation details as a footnote at the end of in Table S2.

9. Line 241 missed a period after the word crucial.

Thank you. We have corrected this in our revised manuscript.

Reference:

- 1 Cajander, S. *et al.* Profiling the dysregulated immune response in sepsis: overcoming challenges to achieve the goal of precision medicine. *Lancet Respir Med* **12**, 305-322, doi:10.1016/S2213-2600(23)00330-2 (2024).
- 2 Shankar-Hari, M. *et al.* Reframing sepsis immunobiology for translation: towards informative subtyping and targeted immunomodulatory therapies. *Lancet Respir Med* **12**, 323-336, doi:10.1016/S2213-2600(23)00468-X (2024).
- 3 Amarasinghe, H. E. *et al.* Mapping the epigenomic landscape of human monocytes following innate immune activation reveals context-specific mechanisms driving endotoxin tolerance. *BMC genomics* **24**, 595, doi:10.1186/s12864-023-09663-0 (2023).
- 4 Fairfax, B. P. *et al.* Innate immune activity conditions the effect of regulatory variants upon monocyte gene expression. *Science* **343**, 1246949, doi:10.1126/science.1246949 (2014).
- 5 Gray, V. *et al.* Hyperglycemia-triggered lipid peroxidation destabilizes STAT4 and impairs anti-viral Th1 responses in type 2 diabetes. *Cell Metab* **36**, 2511-2527 e2517, doi:10.1016/j.cmet.2024.10.004 (2024).
- 6 Zhang, E., Neugebauer, M. E., Krasnow, N. A. & Liu, D. R. Phage-assisted evolution of highly active cytosine base editors with enhanced selectivity and minimal sequence context preference. *Nature communications* **15**, 1697, doi:10.1038/s41467-024-45969-7 (2024).
- 7 Winer, S. *et al.* Normalization of obesity-associated insulin resistance through immunotherapy. *Nature medicine* **15**, 921-929, doi:10.1038/nm.2001 (2009).
- 8 Rocha, V. Z. *et al.* Interferon-gamma, a Th1 cytokine, regulates fat inflammation: a role for adaptive immunity in obesity. *Circulation research* **103**, 467-476, doi:10.1161/CIRCRESAHA.108.177105 (2008).
- 9 Roquilly, A. *et al.* Interferon gamma-1b for the prevention of hospital-acquired pneumonia in critically ill patients: a phase 2, placebo-controlled randomized clinical trial. *Intensive Care Med* **49**, 530-544, doi:10.1007/s00134-023-07065-0 (2023).
- 10 Kwok, A. J. *et al.* Neutrophils and emergency granulopoiesis drive immune suppression and an extreme response endotype during sepsis. *Nature immunology* **24**, 767-779, doi:10.1038/s41590-023-01490-5 (2023).

REVIEWER COMMENTS

Reviewer #1 (Remarks to the Author):

The Authors have extensively revised the manuscript, demonstrating scientific rigor and attention to each point raised. The methodology is convincing, and the point-to-point responses are comprehensive and well justified. Both the strengths and the corresponding limitations of the manuscript are clearly articulated. The revised version further enhances the quality and general impact of a very intriguing study.

Thank you.

Reviewer #2 (Remarks to the Author):

This is a resubmission of a very interesting manuscript, which is highly clinically significant to the field. The work supports the conclusions, and adequate evidence is provided. The methodology is sound. The authors have answered all the queries and changed the manuscript accordingly.

Thank you.

Reviewer #3 (Remarks to the Author):

Follow-up comment (on Major Comment)

I appreciate the authors' additional data on neutrophil viability and their clarification regarding the co-culture experiments. However, I remain concerned about the experimental design and the biological relevance of the time points used. First, the inclusion of day 4 neutrophil data is not appropriate, as neutrophils are extremely short-lived cells that rarely maintain functionality beyond 24 hours *ex vivo*. The authors themselves report only ~25% viability at day 4, indicating that the vast majority of cells are dead or undergoing apoptosis/NETosis. Data derived from such cultures are unlikely to represent true neutrophil-T cell interactions and should therefore be removed entirely from the manuscript and supplementary materials. Second, even at day 2, viability is only ~70–75%, which remains relatively low for reliable interpretation of immune crosstalk. At this stage, dying cells and NETotic debris can substantially affect cytokine levels, activation markers, and readouts attributed to "functional" interactions.

To strengthen the physiological relevance of these assays, I strongly recommend that the authors consider performing shorter co-cultures (e.g., 12–18 hours), which would better reflect viable neutrophil-T cell communication. Such experiments would help confirm whether the observed effects truly arise from active crosstalk rather than secondary consequences of neutrophil death.

If shorter co-cultures cannot be performed at this stage, the authors should explicitly acknowledge this limitation in the Discussion and state clearly that results obtained at or beyond 2 days must be interpreted with caution due to progressive neutrophil apoptosis and loss of function.

We thank the reviewer for these comments and the opportunity to clarify our experimental design.

As indicated in the Methods and figure legends, all co-culture experiments testing Neu-T cell interactions in this study were performed for 2 days. The 4-day rapamycin treatment shown in **Supplementary Fig.5c** and **Supplementary Fig.5c-f** were performed solely to assess dose and time-dependent effects of rapamycin and its potential cytotoxicity, as addressed previously. These 4-day data are not used to interpret neutrophil-T cell crosstalk and are included only for completeness regarding rapamycin exposure and neutrophil viability.

Regarding the reviewer's concern about neutrophil viability, as shown in **Fig.4a**, the inhibitory effect of sepsis neutrophils on T cells is activation-dependent, requires direct cell-cell contact, and is partially reversible according to previously reported mechanisms (Kwok AJ *et al. Nat Immunol.* 2023; PMID: 37095375). These features support the conclusion that the observed effects are due to active interactions rather than nonspecific consequences of cell death or NETotic debris. For the co-culture experiments examining reciprocal Neu-T crosstalk, the 2-day culture interval was selected because it maintains high neutrophil viability while still allowing robust cytokine production from anti-CD3/CD28-stimulated primary T cells. Manufacturer-validated protocols for anti-CD3/CD28 bead activation (Thermo Fisher Dynabeads Human T-Activator CD3/CD28 User Guide) recommend 1-3 days of incubation to achieve sufficient T-cell activation and cytokine secretion. Consistent with this, we observed cytokine-dependent activation of neutrophils in **Fig.4b**, an effect that was not reproduced using Jurkat cells under identical stimulation conditions (**Supplementary Fig.4c-d**). We have now added text in the Discussion section (**line 354-358; page 14**): *"For the co-culture experiments examining reciprocal Neu-T crosstalk, the 2-day culture interval was selected because it maintains high neutrophil viability while still allowing robust cytokine production from anti-CD3/CD28-stimulated primary T cells. Progressive neutrophil apoptosis and loss of function with time is an important potential confounder in such experiments."*

Reviewer #4 (Remarks to the Author):

I have read the revised manuscript and believe the authors have done an excellent job addressing the reviewers concerns. I have no further comments.

Thank you.

Reviewer #5 (Remarks to the Author):

The authors have addressed my previous questions, and I have no additional comments.

Thank you.